# Plasticity in gustatory and nociceptive neurons controls decision making in *C. elegans* salt navigation

Martijn P. J. Dekkers [1,3,4], Felix Salfelder[2,4], Tom Sanders[2], Oluwatoroti Umuerri[1], Netta Cohen [2✉] & Gert Jansen [1✉]

A conventional understanding of perception assigns sensory organs the role of capturing the environment. Better sensors result in more accurate encoding of stimuli, allowing for cognitive processing downstream. Here we show that plasticity in sensory neurons mediates a behavioral switch in *C. elegans* between attraction to NaCl in naïve animals and avoidance of NaCl in preconditioned animals, called gustatory plasticity. $Ca^{2+}$ imaging in ASE and ASH NaCl sensing neurons reveals multiple cell-autonomous and distributed circuit adaptation mechanisms. A computational model quantitatively accounts for observed behaviors and reveals roles for sensory neurons in the control and modulation of motor behaviors, decision making and navigational strategy. Sensory adaptation dynamically alters the encoding of the environment. Rather than encoding the stimulus directly, therefore, we propose that these *C. elegans* sensors dynamically encode a context-dependent value of the stimulus. Our results demonstrate how adaptive sensory computation can directly control an animal's behavioral state.

[1] Department of Cell Biology, Erasmus University Medical Centre, Rotterdam, the Netherlands. [2] School of Computing, University of Leeds, Leeds, UK. [3]Present address: REHAB Basel, Basel, Switzerland. [4]These authors contributed equally: Martijn P. J. Dekkers, Felix Salfelder. ✉email: n.cohen@leeds.ac.uk; g.jansen@erasmusmc.nl

Decision-making refers to the process of choosing among distinct actions as a function of the estimated value of their consequences. In expected utility theory, rational agents assign a subjective value (or expected utility) to a particular action[1]. The subjective value of different outcomes may be context-dependent (e.g., the perceived value of food may be hunger-dependent, as is the cost of food deprivation[2,3]) and limited by noise or partial information[1]. In fact, in all but the simplest behaviors, the information available to an individual will not directly determine the actual utility of different choices to the individual, implying that in the real world, people and animals need to infer, learn, and dynamically adapt the estimated value of different actions. Thus, adaptation is a universal defining feature of animal behavior[4].

Adaptive behavior refers to the ability of animals to change their actions in response to changes in the environment or in their internal state. Here, we study a form of short-term sensory adaptation, called gustatory plasticity[2,5,6]. Its defining feature in the nematode Caenorhabditis elegans is the dynamic balance of salt (NaCl) attraction and avoidance as a function of experience. Often, the goal of behavior is implicit (e.g., tracking an animal and recording its neural activity may not disclose what form of reward the animal seeks as it acts in its environment). Furthermore, the advantage gained by adaptive behavior may be similarly elusive. In fact, as NaCl has little or no objective value to the animal, one might surmise that the expected reward of seeking salt is food. However, information about salt does not directly or reliably predict the presence of food. Gustatory plasticity may therefore be an adaptive mechanism for modulating the expected utility of following (or avoiding) salt concentration gradients, in search of food.

The response of C. elegans to NaCl is associated with food. Naïve animals, cultured in the presence of food and NaCl, will move up NaCl concentration gradients in search of bacteria[2,7,8]. Preconditioned animals, exposed for 15 min to 100 mM NaCl in the absence of food, avoid any NaCl concentration. This switch from attractive to aversive NaCl behavior is called gustatory plasticity[2,5,6]. Gustatory plasticity is reversible, lasting less than 5 min[6]. Thirty minutes or longer exposure to NaCl in the absence of food induces stronger avoidance responses that rely on mostly independent mechanisms[5,9–13].

While adaptive decision-making typically involves an integration of sensory stimuli with information about prior experience and internal state, the neural basis for adaptive decision making is still far from understood. Most studies of adaptive behavior have focused on sites of multi-sensory and internal-state integration in downstream neurons and circuits[14]. We present evidence for history-dependent modulation of value encoding and decision making by sensory neurons during gustatory plasticity in the nematode C. elegans.

Naïve C. elegans are attracted to NaCl concentrations of up to 200 mM but avoid higher NaCl concentrations[2,7,8]. Attraction to NaCl is primarily mediated by the bilaterally asymmetric ASE sensory neurons[15,16]. The left ASE neuron (ASEL) produces Ca²⁺ transients in response to increases in NaCl concentration[17]; the right neuron (ASER) responds to decreases in NaCl concentration[17]. Avoidance of dangerously high NaCl concentrations is mediated by the ASH neurons[7,18]. Previous studies have shown that the ASE and the ASH neurons are involved in gustatory plasticity[2]. In addition, many signaling proteins involved have been identified, including serotonin, dopamine, glutamate, and neuropeptide neurotransmission[2,5,6,9]. In this paper, we study adaptive mechanisms in ASEL, ASER, and ASH to identify possible neuronal and circuit mechanisms of gustatory plasticity and to link neuronal dynamics with the animal's adaptive behavior.

Cell-specific Ca²⁺ imaging in awake animals identified three distinct forms of adaptation that occur in the absence of food and overlap with the timescale of gustatory plasticity: ASEL desensitization upon exposure to NaCl; ASER sensitization to NaCl; and ASH sensitization to considerably lower (non-toxic) levels of NaCl. An additional, fast form of dynamic-range adaptation is identified in ASE sensory neurons, resulting in a logarithmic response amplitude to changes of NaCl concentration, analogous to the Weber–Fechner law of sensory perception[19,20]. Using computational models, we identify a hierarchy of molecular, cellular, and distributed circuit mechanisms that capture our Ca²⁺ imaging results in sensory neurons. Simulations of model animals in a virtual assay environment captured the behavioral switch from attraction to avoidance in gustatory plasticity.

Our experimental results and computational model point to a number of predictions: First, ASH sensitization is necessary and sufficient to explain the behavioral switch in gustatory plasticity. Second, bilateral asymmetries in ASE adaptation limit the animals' ability to follow NaCl gradients but make these neurons excellent adaptive encoders of context- and history-dependent value that drives different motor actions on different timescales. Finally, we postulate a role of sensory adaptation in setting the balance of exploration and exploitation in ecologically relevant scenarios and use our computational framework to support this conjecture in a simplified virtual assay.

## Results

**Naïve sensory responses to NaCl.** What drives the behavioral switch between NaCl attraction and avoidance during gustatory plasticity? Before addressing this question, we determined the range of the naïve responses to NaCl of the ASEL, ASER, and ASH neurons, using the Ca²⁺ reporter Yellow Cameleon[21,22]. Similar to previous findings[17] ASEL neurons produced Ca²⁺ transients in response to a 3 s exposure to both low and high NaCl concentrations, with strongest responses to 200 mM NaCl (Fig. 1a, b; Supplementary Table 1). ASH neurons are known to yield Ca²⁺ transients in response to osmotic stimuli[23]. We recorded Ca²⁺ transients in ASH neurons in response to a 3 s exposure to various NaCl concentrations. We found a gradual increase in the fraction of animals that responded (depicted as the response index, RI, Supplementary Table 1) and in the amplitude of Ca²⁺ transients with increasing concentrations of NaCl, resulting in strong Ca²⁺ fluxes in response to 300 mM and 500 mM NaCl, but only a small fraction of animals responded to 100 or 200 mM NaCl and the associated Ca²⁺ fluxes were weak (Fig. 1d,e; Supplementary Table 1).

**Prolonged exposure to NaCl sensitizes ASER.** In contrast to previous studies which have found ASER responses to decreases in NaCl concentrations[17,24,25], we did not find responses to NaCl concentration decrease after a 3 s exposure, either at low or at high concentrations (Fig. 1c; Supplementary Table 1). This surprising result, combined with the fact that the ASER neuron is known to contribute to NaCl chemotaxis[17,24,25], led us to conjecture that ASER responses may depend on its history of exposure to NaCl. To test this hypothesis, we measured ASER Ca²⁺ responses in animals exposed to 100 mM NaCl for 30 s to 10 min. We found that the fraction of animals that responded, as well as the amplitude of the response increased with exposure time (Fig. 2a, b; Supplementary Table 1), indicating that ASER is gradually sensitized by prolonged exposure to NaCl and confirming that its responses are consistent with positive (attractive) chemotaxis over behaviorally relevant timescales[25,26].

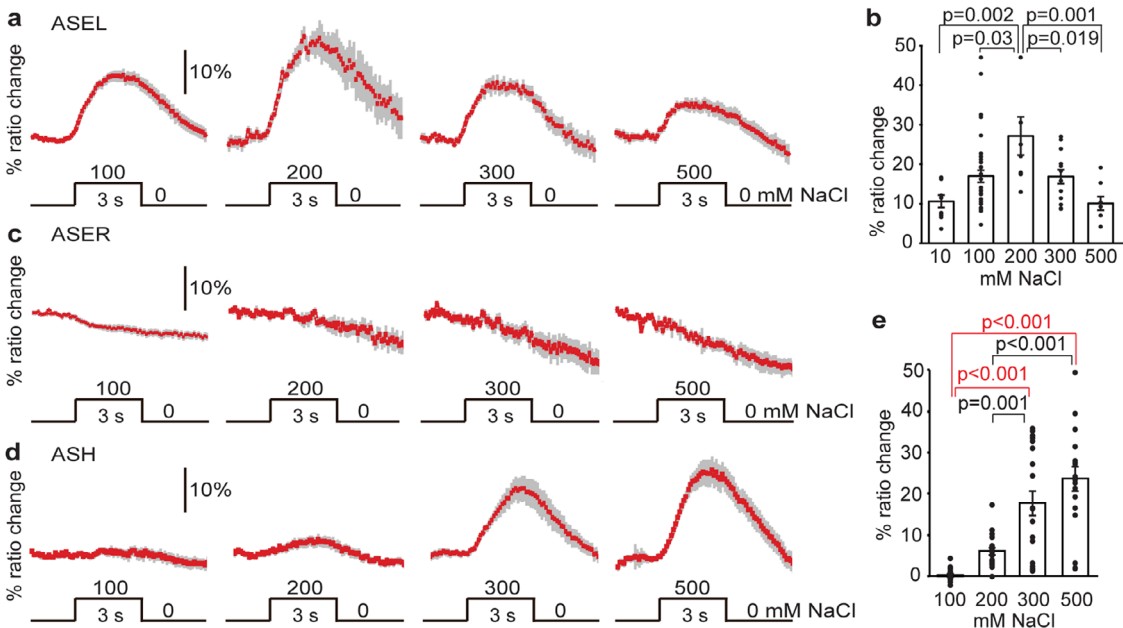

**Fig. 1 Ca$^{2+}$ responses of ASEL, ASER, and ASH neurons to brief NaCl exposure.** Animals were exposed for 3 s to different concentrations of NaCl from a baseline of 0 mM. **a** Average Ca$^{2+}$ transient (±SEM in gray) in ASEL in response to 100–500 mM NaCl; 100 mM: $n = 37$, 200 mM: $n = 9$, 300 mM: $n = 15$, 500 mM: $n = 8$ animals. **b** Average maximum ratio changes (±SEM) in ASEL: responses to 10 ($n = 10$ animals), 100 ($n = 37$), 300 ($n = 15$) and 500 ($n = 8$) mM were significantly different from the response to 200 mM ($n = 9$). **c** Average Ca$^{2+}$ transients (±SEM in gray) in ASER in response to 100–500 mM NaCl; 100 mM: $n = 30$, 200 mM: $n = 5$, 300 mM: $n = 9$, 500 mM: $n = 4$ animals. No statistically significant differences were observed ($p > 0.05$). **d** Average Ca$^{2+}$ transients (±SEM) in ASH after exposure to 100–500 mM NaCl; 100 mM: $n = 18$, 200 mM: $n = 18$, 300 mM: $n = 22$, 500 mM: $n = 19$ animals. **e** Average maximum ratio changes (±SEM) in ASH: responses to 300 and 500 mM were significantly different from the responses to 100 and 200 mM. Traces indicate average percentage change in $R/R_o$ where $R$ is the fluorescence emission ratio and $R_o$ is the baseline fluorescence emission ratio before exposure to NaCl. Individual data points have been indicated as dots. Statistically significant differences have been indicated (non-significant differences, $p > 0.05$, have not been indicated). Source data underlying this figure are available in Supplementary Data 1.

**Prolonged exposure to NaCl desensitizes ASEL.** To determine whether ASEL responses are also modulated by pre-exposure, we tested ASEL responses to 100 mM NaCl after a period of pre-exposure. Animals were pre-exposed to 100 mM NaCl for periods ranging from 60 s to 10 min, followed by a 60 s wash (Fig. 3a). Ca$^{2+}$ responses in ASEL neurons were strongly reduced or even abolished after 5 or 10 min of pre-exposure to 100 mM NaCl, but unaffected after 1 or 2 min of pre-exposure (Fig. 3a, b). These results correlate well with behavioral assays that showed reduced attraction to NaCl with increasing pre-exposure times (Supplementary Fig. 1), as reported previously for the response to sodium acetate[6].

We further found that ASEL continued to respond to NaCl concentrations above the pre-exposure concentration, e.g., to 300 and 400 mM, but not to 200 mM NaCl (Fig. 3c, d). This finding is in accordance with behavioral data that showed that NaCl pre-exposure strongly affected attraction to lower or similar NaCl concentrations but had less or no effect on higher NaCl concentrations (Supplementary Fig. 2), suggesting that ASEL desensitization involves threshold modulation.

Adaptation in ASEL is easily reversible, as washing for 2 or 5 min after 10 min of pre-exposure restored reliable Ca$^{2+}$ transients in ASEL (Fig. 3e, f; Supplementary Table 1). This recovery of the response in ASEL is consistent with behavioral data, where attraction to NaCl after pre-exposure is restored by a 5-min wash[6].

We conclude that the ASEL neuron desensitizes with pre-exposure to NaCl and recovers in the absence of NaCl exposure and suggest that this sensory adaptation modulates the strength of attraction to NaCl in behavioral assays.

**Prolonged exposure to NaCl sensitizes ASH.** To test if the response of the ASH neurons is affected by pre-exposure, we first exposed animals to 100 mM NaCl for 10 min and subsequently introduced a further 100 mM increase to 200 mM NaCl. Strikingly, 16 of the 17 pre-exposed animals (RI 0.94) responded to 200 mM NaCl after pre-exposure, whereas only 5 of 18 animals (RI 0.28) had responded to 200 mM without pre-exposure (Fig. 4a–c; Supplementary Table 1). Response rates and amplitudes to this pre-exposure-stimulus combination were comparable to naïve responses to 500 mM NaCl (Fig. 4a–c; Supplementary Table 1). Thus, ASH neurons are sensitized by 10 min pre-exposure to 100 mM NaCl, upon which they show robust responses to 200 mM NaCl, or to an increase of 100 mM NaCl. Pre-exposure affected neither the number of animals that responded nor the amplitudes of the Ca$^{2+}$ transients in ASH neurons upon exposure to 300 or 500 mM NaCl (Fig. 4a, c; Supplementary Table 1).

**Desensitization of ASEL is likely cell-autonomous.** Top-down modulation of sensory responses is prevalent in many nervous systems, including in that of *C. elegans*[3]. To determine whether desensitization of ASEL requires input from other neurons, we recorded the Ca$^{2+}$ responses in ASEL neurons of mutants previously shown to affect gustatory plasticity in specific sensory neurons. We tested a mutant in the G protein α subunit *odr-3* that functions in gustatory plasticity in the ADF neurons, serotonergic neurons that play a role in dauer formation, and a minor role in chemotaxis to NaCl[2,15,27]. In addition, we tested a mutant in the G protein γ subunit *gpc-1* that functions in gustatory plasticity in the ASI and ASH neurons[6]. ASI neurons are also involved in dauer formation and have a minor role in chemotaxis to NaCl[15]. Furthermore, we tested animals that overexpress the *lsy-6* gene in both ASE neurons, resulting in the transformation of the ASER neuron to an ASEL neuron[28]. Finally, we tested *unc-13(e51)* mutants with

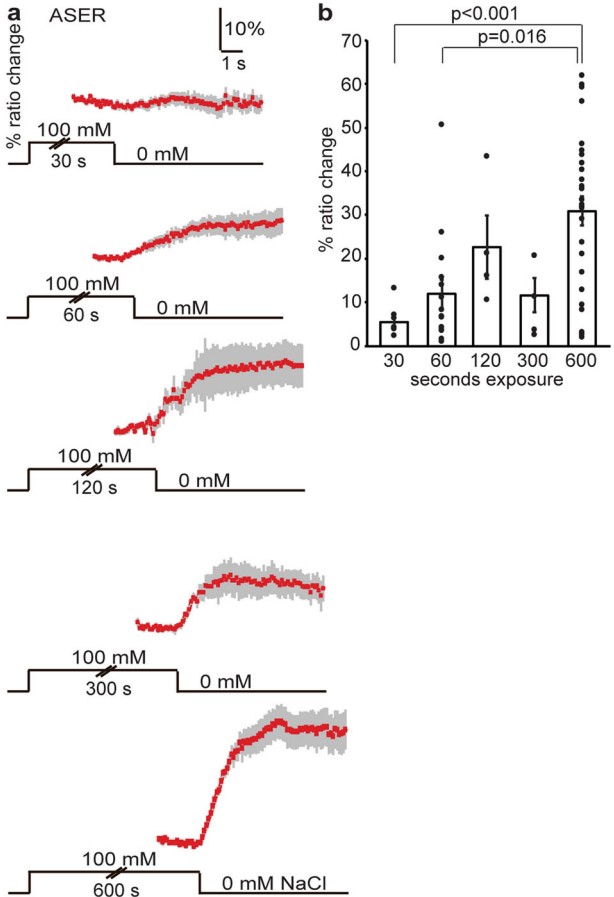

**Fig. 2 Prolonged exposure to NaCl sensitizes ASER. a** Average $Ca^{2+}$ transient (±SEM) in ASER in response to a decrease in NaCl concentration from 100 mM to 0 mM after 30–600 s exposure. Thirty seconds of exposure to 100 mM NaCl did not result in a response in ASER, but longer exposures did. **b** Average maximum ratio changes (±SEM) in ASER after 30–600 s exposure. Thirty seconds: $n = 8$, 1 min: $n = 15$, 2 min: $n = 4$, 5 min: $n = 5$, 10 min: $n = 29$ animals. Individual data points have been indicated as dots. Statistically significant differences have been indicated (non-significant differences, $p > 0.05$, have not been indicated). Source data underlying this figure are available in Supplementary Data 2.

disrupted synaptic vesicle release, *eat-4(ad819)* mutants with defective vesicular glutamate transport, *unc-31(e928)* and *egl-3(ok979)* mutants with neuropeptide signaling defects, *cat-2(tm2261)* animals with defective dopamine synthesis and *tph-1(mg280)* mutants, which fail to produce serotonin[27,29–33]. Interestingly, none of these mutants showed a significant reduction in desensitization (Supplementary Fig. 3), suggesting that desensitization of ASEL is cell-autonomous. Only *tph-1* mutants showed slightly abnormal ASEL desensitization; five out of nine animals tested showed a weak response to 100 mM NaCl after 10 min pre-exposure, whereas none of the 16 wild-type animals tested responded (Supplementary Fig. 3c). However, as the average maximum ratio change in pre-exposed *tph-1* animals was not statistically different from that of wild-type animals, further analyses are required to reveal a possible contribution of serotonin to ASEL desensitization.

Taken together, our results suggest cell-autonomous desensitization of the ASEL neuron after prolonged exposure to NaCl.

**Sensitization of ASER is likely cell-autonomous**. To determine whether sensitization of ASER requires neuropeptide, dopamine,

or serotonin signaling, we tested the response of the ASER neuron of *egl-3(pk979)*, *cat-2(tm2261)*, and *tph-1(mg280)* mutant animals. Mutations in these genes did not affect ASER sensitization (Supplementary Fig. 4).

In agreement with previous data[17], we found a strong response of the ASER neuron of *unc-13(e51)* animals to a decrease in NaCl after 10 min exposure, similar to wild-type animals (Fig. 5a, b), indicating that sensitization of ASER in response to NaCl pre-exposure does not require synaptic neurotransmission. However, unlike the wild type, 67% of *unc-13(e51)* animals also responded to a decrease in NaCl after 30 s of exposure, resulting in a slow rise of $Ca^{2+}$ (Fig. 5a–c). Although these data have not been confirmed in a second *unc-13* mutant strain or by a rescue experiment, our findings suggest that the response of ASER is inhibited by a weak synaptic signal that abolishes the graded depolarization of the cell in naïve (desensitized) animals.

Finally, *eat-4(ad819)* mutant animals did not respond to a decrease in NaCl after 30 s exposure but did respond after 10 min of exposure, indicating that glutamate signaling is not required for ASER sensitization (Fig. 5a, d). However, the onset of the ASER response in *eat-4* mutant animals was ~2 s delayed, compared to the almost immediate response of ASER in wild-type animals (Fig. 5a, d). Thus, although these findings have not been confirmed by a rescue experiment, glutamate seems to be involved in facilitating a rapid onset of a $Ca^{2+}$ response to a decrease in NaCl. Since no delay in ASER response was observed in *unc-13(e51)* mutant animals, the glutamate signal might be extra-synaptic in origin. Further experiments are required to reveal the nature of this signal.

Taken together, we conclude that sensitization of the ASER neuron in response to prolonged exposure to NaCl is likely cell-autonomous. We also found that the ASER $Ca^{2+}$ response likely involves a glutamate-mediated signal that advances the onset of the response, and a synaptic signal that abolishes a weak graded response in naïve animals, neither of which are required for gustatory plasticity.

**ASH sensitization requires ASE, glutamate, neuropeptides, dopamine, and serotonin**. Since ASE neurons are required for gustatory plasticity[2] we tested whether they are required for sensitization of ASH by measuring ASH responses in *che-1* mutants that lack functional ASE neurons[34]. *che-1* mutants were indistinguishable from wild-type animals in their naïve response to 200 or 500 mM NaCl (Fig. 6c, e, compared to wild-type naïve responses in Fig. 1d, e), but failed to respond or responded very weakly to 200 mM NaCl following pre-exposure to 100 mM NaCl for 10 min (Fig. 6a, d, e). To control for the responsiveness of the ASH neurons of the tested animals, we confirmed the response of the same animals to 500 mM NaCl (Fig. 6d). These data show that ASH sensitization is a circuit effect that requires the ASE neurons. Whether ASE neurons recruit ASH by allowing it to sense NaCl at lower concentrations (akin to a form of threshold modulation), or whether an ASEL/R sensory signal is transmitted to ASH that effectively acts as an interneuron, reminiscent of AWC recruitment by ASE[35], remains unknown.

We next asked which neurotransmitters play a role in ASH sensitization. Interestingly, all mutants tested, *unc-13(e51)*, *eat-4(ad819)*, *unc-31(e928)*, *egl-3(ok979)*, *cat-2(tm2261)*, and *tph-1(mg280)*, showed reduced sensitization of ASH, but responded as wild type to 500 mM NaCl, suggesting that synaptic transmission, glutamate, neuropeptides, dopamine, and serotonin signaling all play a role in sensitization of ASH (Fig. 6f–k). Previously, Hilliard *et al.* have shown that mutation of *unc-13* or incubation in serotonin does not affect the $Ca^{2+}$ response of ASH to an osmotic stimulus[23].

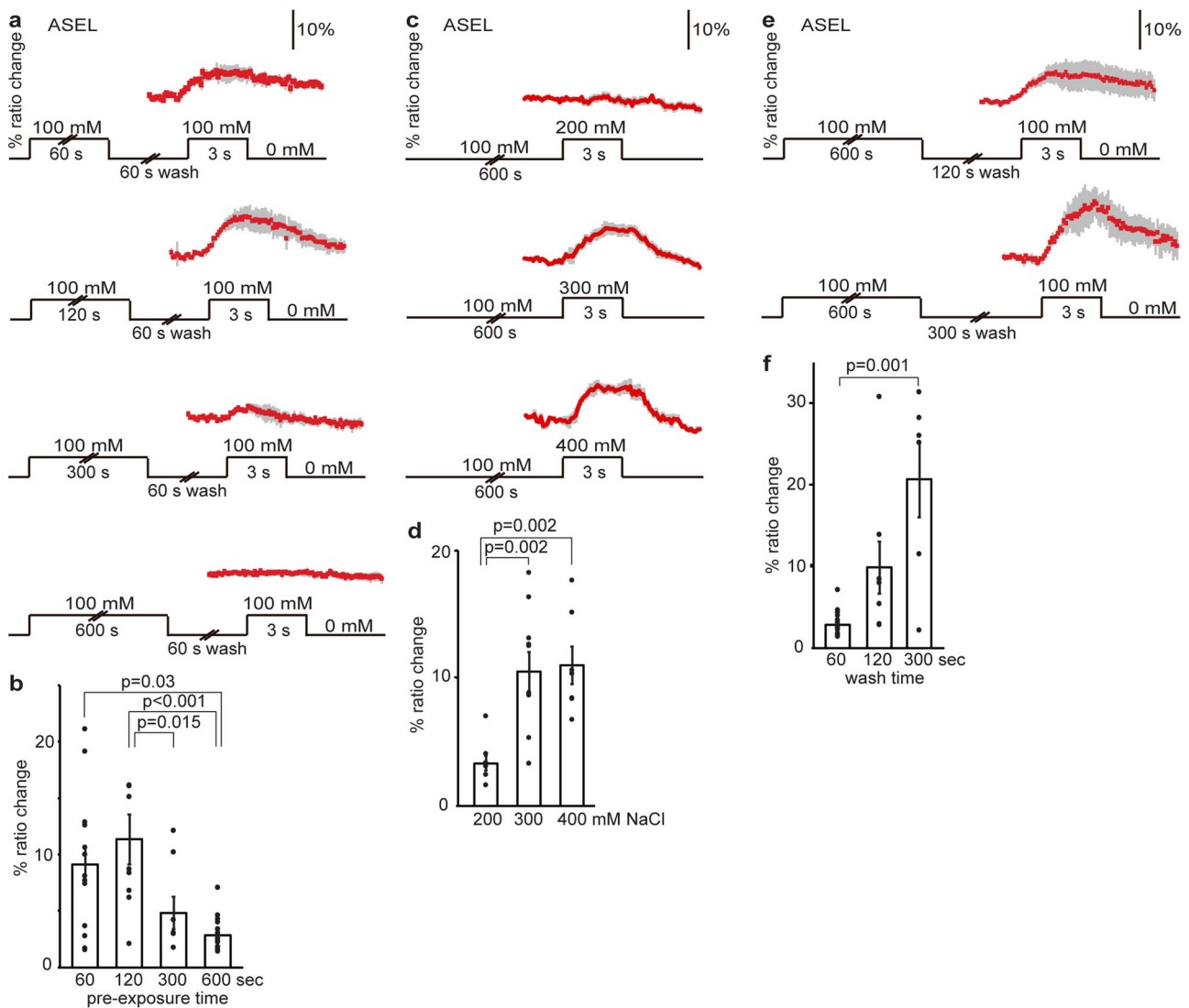

**Fig. 3 Pre-exposure to NaCl desensitizes ASEL. a, b** Animals were pre-exposed to 100 mM NaCl for 60–600 s, washed briefly (60 s), and exposed to 100 mM NaCl. **a** Average $Ca^{2+}$ transient (±SEM) in ASEL in response to 100 mM NaCl after pre-exposure for 60 s ($n = 14$), 120 s ($n = 9$), 300 s ($n = 8$), or 600 s ($n = 16$ animals). **b** Average maximum ratio changes (±SEM) in ASEL after pre-exposure: responses after 300 and 600 s pre-exposure were significantly reduced, compared to responses after 60 or 120 s pre-exposure. **c, d** Animals were pre-exposed to 100 mM NaCl for 10 min, before test exposure to 200, 300, or 400 mM NaCl. **c** The average $Ca^{2+}$ transients (±SEM) and (**d**) the average maximum ratio changes (±SEM) in ASEL. Exposure to 300 ($n = 10$ animals) or 400 mM ($n = 7$) NaCl yielded significantly stronger $Ca^{2+}$ transients than 200 mM ($n = 8$). **e, f** Animals were pre-exposed to 100 mM NaCl for 10 min, washed for 1, 2, or 5 min in a NaCl-free buffer and re-exposed to 100 mM NaCl, where (**e**) shows the average $Ca^{2+}$ transients (±SEM) and (**f**) the average maximum ratio changes (±SEM) in ASEL. Five minutes wash with a NaCl-free buffer restored the $Ca^{2+}$ response of ASEL to 100 mM NaCl. Wash time: 1 min ($n = 16$), 2 min ($n = 8$), 5 min ($n = 6$ animals). Individual data points have been indicated as dots. Statistically significant differences have been indicated (non-significant differences, $p > 0.05$, have not been indicated). Source data underlying this figure are available in Supplementary Data 3.

We conclude that the recruitment of ASH to respond to non-toxic levels of NaCl requires ASE and relies on multiple pathways involving synaptic transmission, glutamate, neuropeptides, dopamine, and serotonin signals.

**In silico sensory neurons support fast dynamic range adaptation to NaCl.** To better understand the behavioral implications of the different forms of sensitization and desensitization in ASEL, ASER and ASH neurons, we used our empirical results to construct a computational model (Fig. 7a–d). Many *C. elegans* sensory neurons, including ASE and ASH, respond to the change in stimulus over time[17,36]. Transient pulse-like responses are well captured by two opposing and timescale separated components[37]. In the absence of detailed conductances, we imposed dynamics

that closely mimic models of eukaryotic chemotaxis[38] (Fig. 7a–c; Supplementary methods) by letting the slow variable, hyperpolarizing current, denoted $S$, follow the fast, depolarizing current, $F$, with a delay (Supplementary Fig. 10 in Supplementary methods). The model ASEL depolarizes to NaCl increases, while ASER depolarizes to NaCl decreases and hyperpolarizes to NaCl increases[17]. Since ASHL and ASHR respond identically to NaCl and are electrically coupled, we modeled them as a single unit which depolarizes to NaCl increases (ASH, Fig. 7c)[36,39–41]. As hyperosmotic responses are unaffected by gustatory plasticity, they were not considered in our model.

We parameterized the model sensory cells using $Ca^{2+}$ imaging data from fully sensitized ASEL and ASER neurons. The rise times, and even more so, the decay times of the responses were

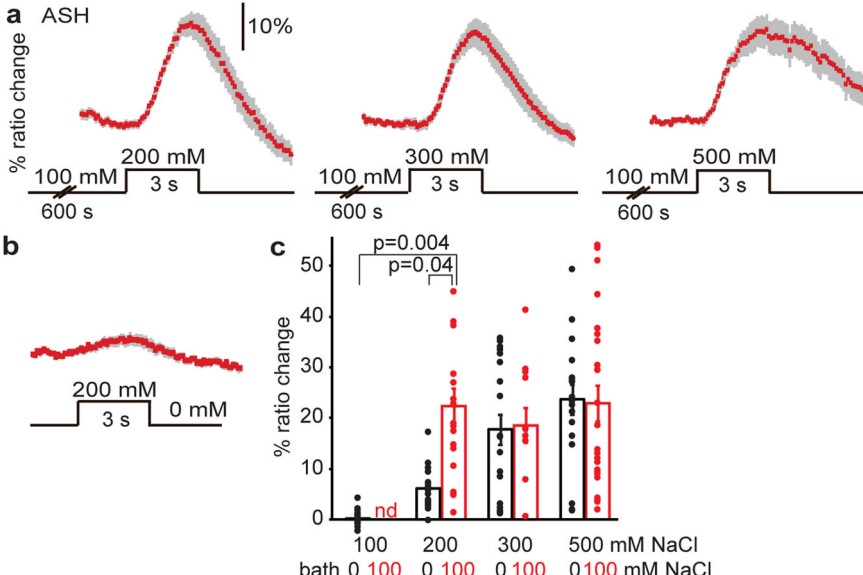

**Fig. 4 Prolonged exposure to NaCl sensitizes ASH. a** Average $Ca^{2+}$ transient (±SEM) in ASH in response to an increase from 100 mM NaCl (after 600 s exposure) to 200–500 mM NaCl. The response of ASH to 200 mM NaCl was increased after pre-exposure to 100 mM NaCl while the responses to 300 and 500 mM NaCl were unchanged. **b** Average $Ca^{2+}$ transients (±SEM) in ASH after exposure to 200 mM NaCl from a baseline of 0 mM NaCl (data from Fig. 1). **c** Average maximum ratio changes (±SEM) in ASH after exposure to 100, 200, 300, or 500 mM NaCl, in animals pre-exposed to 100 mM NaCl for 600 s, or to control condition (100 or 0 mM NaCl bath solution, respectively). Control: 100 mM: $n = 18$, 200 mM: $n = 18$, 300 mM: $n = 22$, 500 mM: $n = 19$ animals. Pre-exposed to 100 mM NaCl: 200 mM: $n = 17$, 300 mM: $n = 12$, 500 mM: $n = 24$ animals. Individual data points have been indicated as dots. Statistically, significant differences have been indicated. Source data underlying this figure are available in Supplementary Data 4.

consistently and significantly faster in ASEL than in ASER, consistent with previous work[17] (Figs. 1a and 2a; Supplementary Table 1; Supplementary Fig. 5; decay times: ASEL responses were back to baseline in 3.9 ± 1.3 s, whereas ASER responses decreased by only ~15% of the maximum amplitude in 6 s).

Our and previous[17] $Ca^{2+}$ imaging results consistently show that the peak depolarization amplitude varies with the stimulus intensity. The attractive chemotaxis responses to NaCl concentrations ranging from 0.1 mM to 100 mM best fit a logarithmic relationship[2,6]. Similarly, we found that a logarithmic function of stimulus intensity best reproduced our ASEL and ASER $Ca^{2+}$ response data, strongly reminiscent of the Weber–Fechner law of sensory perception, which states that the ability to distinguish between two magnitudes of a stimulus scales with the magnitude; a mathematically equivalent form is, $r \propto \log s$, where $r$ and $s$ are the response and stimulus, respectively[3,19]. We chose a parsimonious representation of this dynamic range modulation in which sensory neurons instantaneously respond to the logarithm of the NaCl concentration (Supplementary Methods, Fig. 7a–c). Results showed close agreement with $Ca^{2+}$ traces in ASEL and ASER (Supplementary Fig. 11 in Supplementary methods).

**Gustatory adaptation occurs downstream of dynamic range adaptation.** We next incorporated adaptation into our model sensory neurons, with parameters constrained by our $Ca^{2+}$ imaging data. ASEL was desensitized to stimuli below the concentration of NaCl pre-exposure, but continued to respond to higher concentrations, consistent with threshold adaptation ($C_0$ in Fig. 7a). ASER adaptation was modeled as gain modulation ($D$ in Fig. 7b), consistent with the absence of a response in the naïve context (Fig. 1c). Both ASEL and ASER (de)sensitization had to be applied after logarithmic scaling to reproduce the $Ca^{2+}$ imaging data. Thus, this model constraint suggests that gustatory adaptation occurs downstream of the receptor and of rapid dynamic range adaptation.

Our $Ca^{2+}$ imaging data indicate that ASH is recruited into the low-concentration NaCl sensing circuit upon pre-exposure to NaCl. In the absence of a known mechanism, we modeled this minimalistically as an on/off switch, recruiting and releasing ASH from the gustatory circuit. Switching is governed by a stochastic process with dynamic switching rates dependent on the history of the salt concentration (see Supplementary methods).

**In silico animals reproduce neuronal response and behavior of naïve animals.** To model the behavioral consequences of sensory adaptation, we constructed a full sensory-motor model that could be simulated in a virtual assay arena. As we focused on the sensory system, we chose to use a minimal embodiment and a relatively abstract motor system (Fig. 7d). To explore and navigate their environment, *C. elegans* use a combination of steering, where animals gently turn to reorient, and a biased random walk, in which animals reorient by making sharp turns or pirouettes[16,26]. Both strategies allow animals to migrate along chemical gradients. In our model, point animals moved at a fixed speed of 0.11 mm/s[36] with a dynamic bearing, subject to both steering and pirouettes (see Supplementary methods).

To explore gustatory plasticity in silico, we replicated the quadrant NaCl-choice assay[6,42] in our model. Simulations of 1000 wild-type naïve worms, with ASEL, ASER, and ASH adaptation/ recruitment dynamics, for 10 min of virtual time in the quadrant assay, yielded a similar in silico chemotaxis index to experimental naïve results (wild type in Fig. 7e, h).

**Robustness of chemotaxis is maintained with ASE (de)sensitization in our computational model.** The opposite actions of ASEL and ASER adaptation suggest only one of the ASE pair is fully sensitized at any one time. Thus, we expected a severe performance penalty in our simulations, relative to a model with no adaptation. We found that both models with and without sensory adaptation in ASE quantitatively reproduce the chemotaxis index from the quadrant assay (Fig. 7e; Supplementary

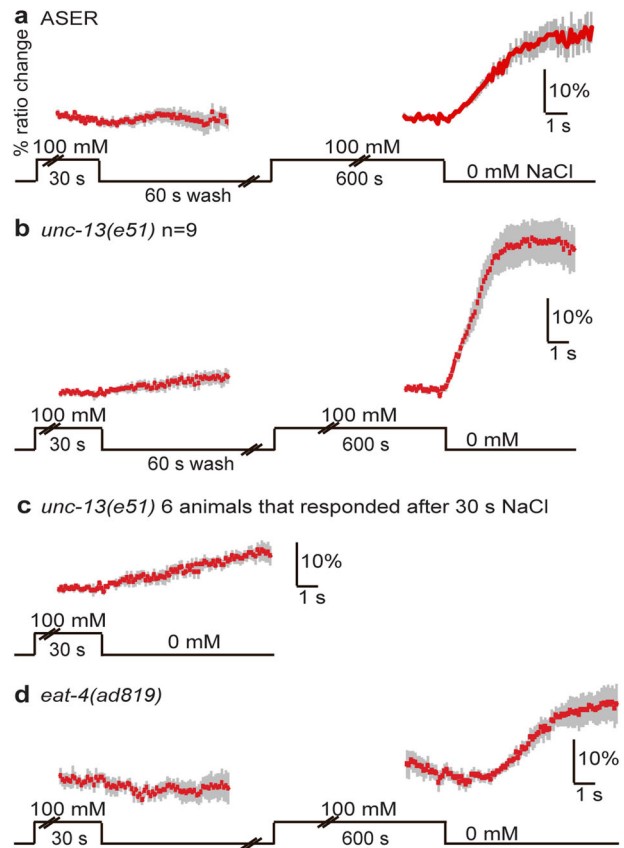

**Fig. 5 ASER sensitization is affected by *unc-13* and *eat-4* mutations. a** Average Ca²⁺ transient (±SEM) in ASER in wild-type animals in response to a decrease in NaCl concentration from 100 mM to 0 mM after 30 and 600 s exposure ($n = 8$ animals). **b** Average Ca²⁺ transients (SEM) in ASER of *unc-13(e51)* animals ($n = 9$) in response to a decrease in NaCl concentration from 100 mM to 0 mM after 30 and 600 s exposure. Thirty seconds of exposure to 100 mM NaCl resulted in a small response in ASER, in six out of nine animals tested. Longer exposure to NaCl resulted in a strong Ca²⁺ response of the ASER neurons of *unc-13* animals. **c** Average Ca²⁺ transients (±SEM) in ASER of the six *unc-13(e51)* animals that responded to a decrease in NaCl concentration from 100 mM to 0 mM after 30 s exposure. **d** Average Ca²⁺ transients (±SEM) in ASER of *eat-4(ad819)* animals ($n = 5$) in response to a decrease in NaCl concentration from 100 mM to 0 mM after 30 or 600 s exposure. Thirty seconds of exposure to 100 mM NaCl did not result in a response. Ten minutes of exposure to 100 mM NaCl did result in a response, albeit 2 s later than in wild-type animals. Source data underlying this figure are available in Supplementary Data 5.

Movies 1 and 2). To better understand the ramifications of adaptation on performance and robustness, we generated variable population models of animals, which we then simulated on our choice assay. Each population consisted of model animals either with (test) or without (control) ASE adaptation. To focus on the role of adaptation, within each population, model animals in their ASE kinetic parameters (determining the rise and decay profiles; see Supplementary Methods). Across a wide range of noise amplitudes, and hence a wide range of sensory neuron parameters, model worms with ASE (de)sensitization performed at least as well as those without sensory adaptation (Supplementary Fig. 7a). Thus, our computational model predicts that the robustness of the performance in the quadrant assay (as measured by the chemotaxis index) is not reduced by ASE (de) sensitization.

**In silico ASH sensitization reproduces gustatory plasticity.** Next, we looked at the behavior of pre-exposed animals focusing on a 15 min 100 mM NaCl pre-exposure, at which the avoidance behavior is the strongest (Supplementary Fig. 2). Since ASER sensitizes, producing stronger attraction, and ASEL desensitizes, producing weaker attraction but not avoidance, ASH seemed a likely candidate to drive salt avoidance. Indeed, to reproduce strong avoidance after pre-exposure, we had to set the synaptic weights of ASH to be stronger than the ASE synaptic weights (Figs. 8a, 10 min; Supplementary Table 2 in Supplementary methods). Such a 'drowning' of attractive signals by ASH is consistent with the strong Ca²⁺ response in ASEL to 300 and 500 mM NaCl in wild-type animals (Fig. 1a) and with behavioral results: While wild-type *C. elegans* are strongly repelled by these concentrations, ASH deficient *odr-3(n1605)* animals are strongly attracted by them[2]. In addition, genetic ablation of ASH strongly reduced avoidance after pre-exposure[2], consistent with the results of our computational model.

**Sensory neuron timing strongly influences navigation strategies in our model.** Our and published Ca²⁺ imaging experiments[17] have revealed a clear timescale separation between the Ca²⁺ responses of fully sensitized ASEL and ASER, both in their rise and decay times (Supplementary Fig. 5). To determine the behavioral consequences of the timescales of ASEL and ASER kinetics, we determined the contributions of ASEL and ASER to steering and pirouettes in our simulations. When we ablated the in silico connection from ASEL to the pirouette motor program or the in silico connection from ASER to the steering circuit, chemotaxis remained unchanged relative to wild-type model animals (Fig. 7f; Supplementary Movies 1, 3–6). Conversely, virtually severing the connection from ASEL to the steering circuit or from ASER to the pirouette motor program severely reduced chemotaxis (Fig. 7f; Supplementary Movies 1, 3–6). Thus, in our model, ASEL controls steering, but has little effect on the pirouette rate, whereas ASER modulates pirouettes, but has little control over steering. We found that disabling steering in simulations of wild-type and ASEL-ablated animals equally reduced chemotaxis in the virtual quadrant assay (Fig. 7f; Supplementary Movies 1, 7–14), confirming this model behavior. Similarly, disabling pirouette modulation in wild-type or ASER ablated animals resulted in equally reduced chemotaxis (Fig. 7f; Supplementary Movies 1, 7–14). Finally, ablating ASEL in animals where pirouette modulation was disabled, or ablating ASER in animals where steering was disabled almost fully abolished the response to NaCl in our model (Fig. 7f; Supplementary Movies 1, 7–14). These results are in agreement with behavioral results of Suzuki et al. who previously showed that ASEL activation promotes runs whereas ASER activation induces turns[17].

Our computational model predicts that both ASEL and ASER contribute to chemotaxis in the quadrant assay. To test this, we genetically ablated either ASEL or ASER, using animals that express Caspase-3 in either the left or right ASE neuron[24]. These animals showed strongly reduced chemotaxis to NaCl (Fig. 7h), confirming that both ASEL and ASER contribute to navigation in the quadrant assay.

To rule out any contribution of desensitization of ASEL and sensitization of ASER to the above analyses, we re-ran our simulations with both ASE neurons fully sensitized and ASH recruitment disabled. These analyses gave results very similar to our previous analyses (Fig. 7g; Supplementary Fig. 7b), confirming that in our model the separate roles of ASEL and ASER in motor control are direct consequences of the rise and decay times of their responses.

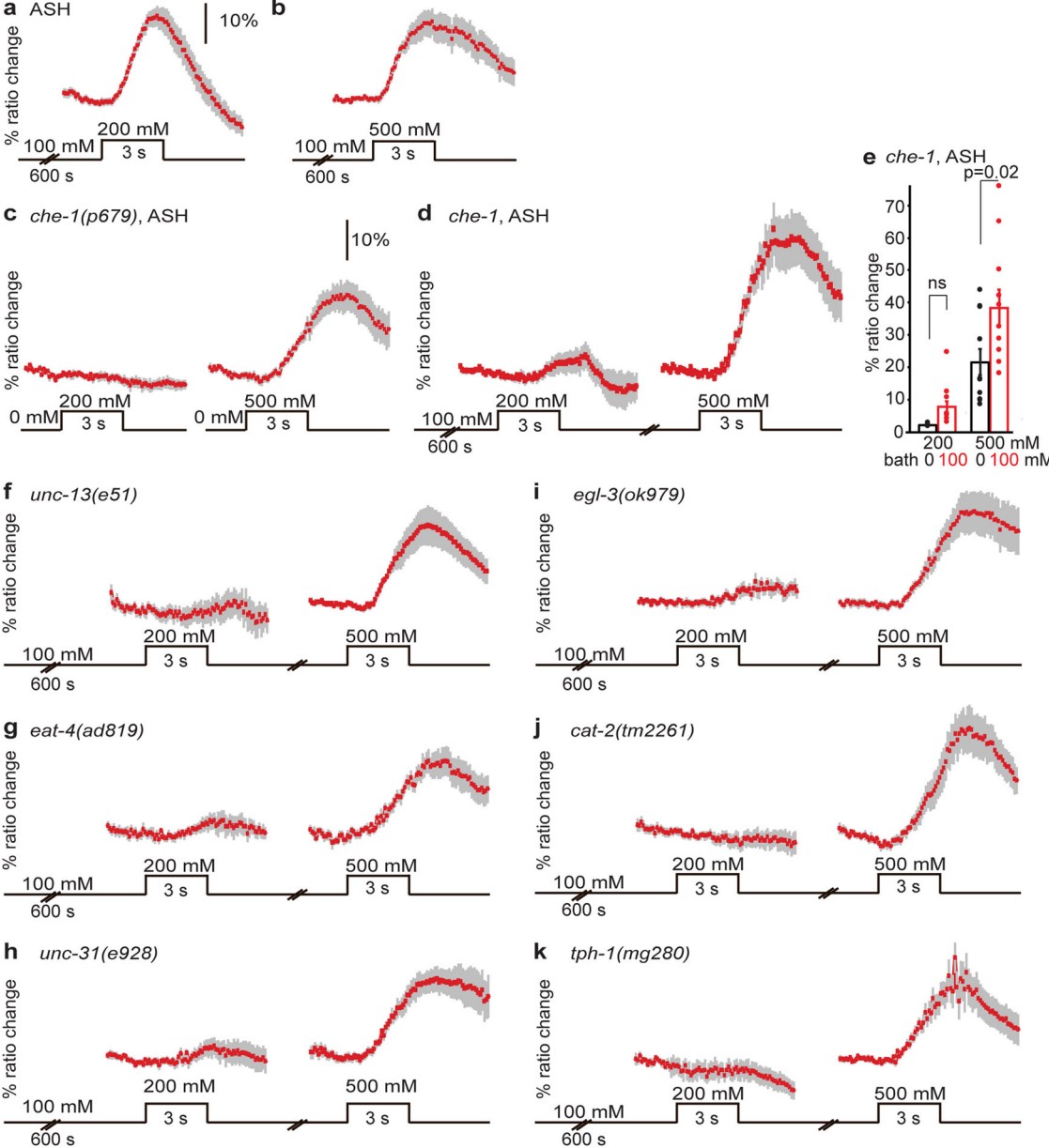

**Fig. 6 ASH sensitization requires ASE, glutamate, neuropeptides, dopamine, and serotonin. a, b** Average Ca²⁺ transients (±SEM) in ASH neurons of wild-type animals after 600 s pre-exposure to 100 mM NaCl in response to an increase to 200 (**a**) or 500 (**b**) mM NaCl (data from Fig. 4). **c** Average Ca²⁺ transients (±SEM) in ASH neurons of *che-1(p679)* animals in response to exposure to 200 or 500 mM NaCl. Naïve *che-1* animals did not respond to 200 mM NaCl (*n* = 6 animals) but did respond to 500 mM (*n* = 11). **d** Average Ca²⁺ transients (±SEM) in ASH neurons of *che-1(p679)* animals after 600 s pre-exposure to 100 mM NaCl in response to an increase to 200 or 500 mM NaCl. Only very weak responses to 200 mM NaCl could be observed in *che-1* animals (*n* = 11, 3 out of 11 worms showed a response). The same *che-1* animals did respond to 500 mM NaCl after a 2 min wash (*n* = 11). **e** Average maximum ratio changes (±SEM) in ASH of *che-1* mutant animals after exposure to 200 or 500 mM NaCl, in animals pre-exposed to 100 mM NaCl for 600 s, or to control condition (100 or 0 mM NaCl bath solution, respectively). Individual data points have been indicated as dots. **f–k** Average Ca²⁺ transients (±SEM) in ASH neurons of various neurotransmitter mutants after 600 s pre-exposure to 100 mM NaCl in response to an increase to 200 and subsequently 500 mM NaCl. None of these mutants showed a significantly stronger response to 200 mM NaCl after pre-exposure than in the naïve situation (*p* > 0.05). Only responses of animals that responded to 500 mM NaCl were included. **f** Two out of eight *unc-13(e51)* animals that responded to 500 mM NaCl responded to 200 mM NaCl after pre-exposure to 100 mM NaCl. **g** Two out of nine *eat-4(ad819)* animals that responded to 500 mM NaCl responded to 200 mM NaCl after pre-exposure to 100 mM NaCl. **h** Three out of seven *unc-31(e928)* animals that responded to 500 mM NaCl responded to 200 mM NaCl after pre-exposure to 100 mM NaCl. **i** Three out of nine *egl-3(ok979)* animals that responded to 500 mM NaCl responded to 200 mM NaCl after pre-exposure to 100 mM NaCl. **j** Three out of nine *cat-2(tm2261)* animals that responded to 500 mM NaCl responded to 200 mM NaCl after pre-exposure to 100 mM NaCl. **k** One out of eight *tph-1(mg280)* animals that responded to 500 mM NaCl responded to 200 mM NaCl after pre-exposure to 100 mM NaCl. Statistically significant differences have been indicated. Source data underlying this figure are available in Supplementary Data 6.

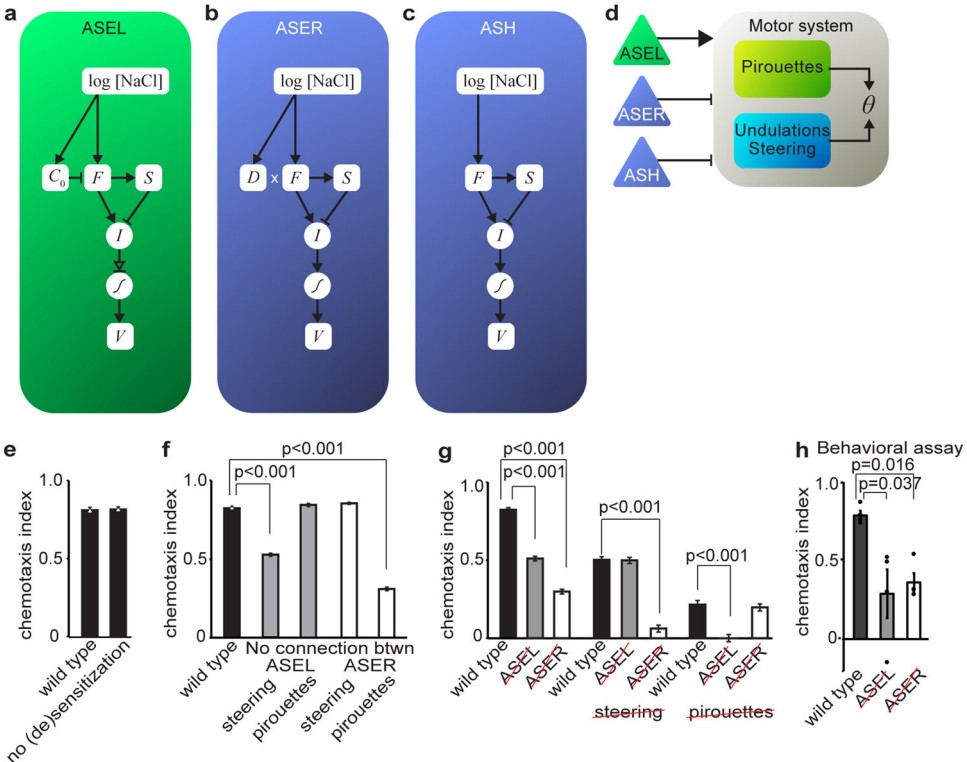

**Fig. 7 Computational model reproduces sensory responses to NaCl in virtual quadrant assay. a–d** Schematic of the computational model. a-c Internal sensory computation in ASEL, ASER and ASH, respectively. Sensory stimuli drive a fast component $F$, and a delayed rectifier $S$, with opposite contributions to the overall current $I$. ASEL desensitization is included as an adaptive threshold $C_0$, ASER sensitization is modeled using a multiplicative gain, $D$, and ASH sensitization is modeled as a stochastic binary switch. The current-voltage relation is given by a sigmoidal activation function. **d** Sensorimotor pathway modulates rhythmic undulations and the frequency of random turns. **e–g** Average chemotaxis index (±SEM) of model wild-type versus mutant animals in the simulated quadrant assay. **e** Our model shows no difference in attraction in the quadrant assay between wild-type model animals ($n = 950$) and animals with always fully sensitized ASEL and ASER (no (de)sensitization, $n = 934$). **f** Ablating the synaptic connections from ASEL to the steering circuit ($n = 962$ animals) or from ASER to the pirouette neuron ($n = 936$) significantly reduced the chemotaxis index (Supplementary Table 2). Ablating the connection from ASER to steering ($n = 967$) very slightly increased the chemotaxis index (wild type: $n = 950$; no connection between ASEL and pirouettes: $n = 952$). **g** Double mutants with ablated ASEL and no steering, or ablated ASER and no pirouettes achieved the same chemotaxis index as single mutants ($n = 938$–$950$ animals). **h** Experimental validation: animals with genetically ablated ASEL (OH8585) and genetically ablated ASER (OH8593) exhibited a strong reduction in the average chemotaxis index (±SEM) in the quadrant assay ($n = 4$ independent assays). Individual data points have been indicated as dots. See corresponding Supplementary Movies 1–14. Statistically, significant differences have been indicated, $*p < 0.05$, $***p < 0.0005$ (non-significant differences, $p > 0.05$, have not been indicated). Source data underlying this figure are available in Supplementary Data 7.

Thus, our model results are consistent with results from ASEL and ASER ablated animals, as well as with observations that both steering and pirouettes contribute to the chemotaxis index in the quadrant assay. In the model, distinct motor programs are separately controlled by ASEL and ASER, as a direct result of the timescales of the sensory neuron responses. To steer, sensory signals must be detected on the timescale of a half-undulation: $O(1–2\,s)$ or faster[26,43,44] (Supplementary Fig. 12 in Supplementary methods). The slower rise time in ASER precludes this (Supplementary Fig. 7b). The contribution of ASER to steering in different assays[26,45] may indicate a faster rise time in ASER. Pirouettes occur with a mean rate of 2.1 events per minute[26] or less (in the quadrant assay). Therefore, to effectively modulate this rate requires a memory of salt exposure over commensurate (or longer) timescales. The fast decay time of ASEL precludes this, while the slow decay time of ASER is ideally suited to modulate the pirouette rate effectively. Should ASER decay on a faster timescale, the modulation of pirouettes would require a slow integration elsewhere in the circuit.

**In silico ASH mediates a detailed balance between attraction and avoidance.** We next asked whether sensory adaptation is

sufficient to account for the balance of attraction and avoidance over time. We therefore followed the long-term behavior of real animals in the quadrant assay over 1 h. For both naïve and pre-exposed animals, the chemotaxis index dropped to approximately zero over the course of the hour, indicating roughly equal numbers of animals in the salt and no salt quadrants (Fig. 8a, green lines). Strikingly, without any further parameter tuning, simulations of naïve and pre-exposed animals closely reproduced these experimental data (Fig. 8a, red and orange lines).

To determine the potential contribution of ASH sensitization to the chemotaxis index decay to zero over time we simulated animals having either ASH completely disabled or fully recruited. Now the chemotaxis index decayed only partially, reaching a plateau around 0.6 and −0.7 respectively (Fig. 8b, blue lines). Conversely, our simulations including ASH sensitization dynamics showed that ASH recruitment inside the NaCl quadrants (driving avoidance) and ASH relaxation outside the NaCl quadrants (allowing attraction) lead to an equal number of animals inside and outside the NaCl quadrants. These results point at a detailed balance description[46] of ASH dynamics, in which animals stochastically switch between recruited and unrecruited ASH states, consistent with the finding that on average ASH dynamics are governed by similar time scales of

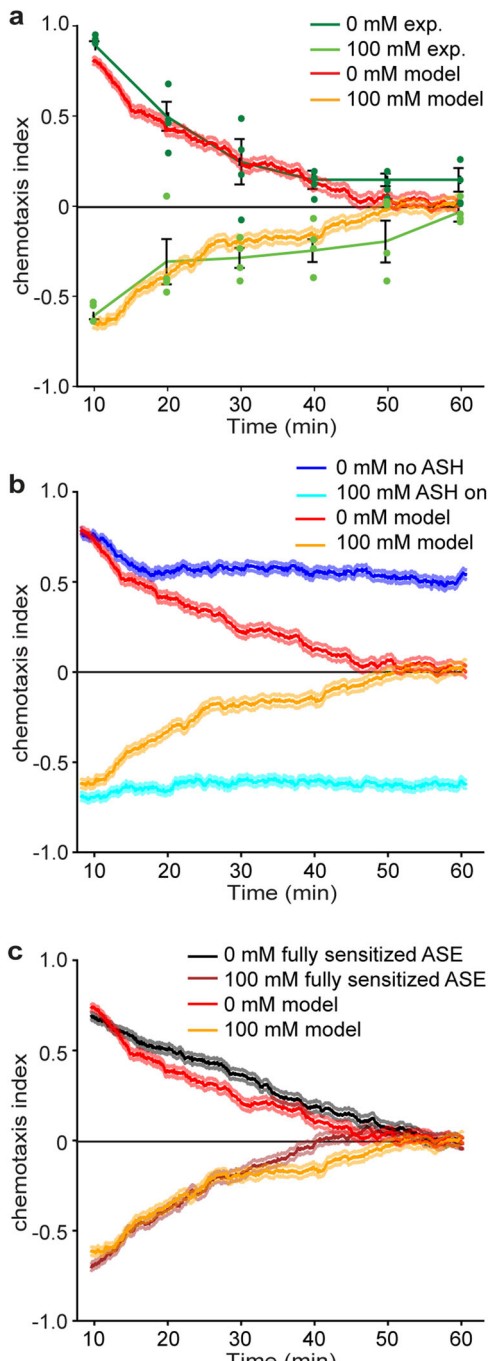

**Fig. 8 Stochastic recruitment of ASH drives gustatory plasticity in our computational model.** Chemotaxis index over time in experiments and simulations of the quadrant assay. **a** Average chemotaxis index (±SEM, individual data points have been indicated as dots) of naïve (0 mM) and pre-exposed (15 min, 100 mM NaCl) animals ($n = 4$ independent assays). The behavioral results (green) and the modeling results (red, orange) show a monotonic decay towards a chemotaxis index of 0, for both naïve and pre-exposed animals. **b** Virtual chemotaxis index for animals with unrecruited ASH (dark blue) and with recruited ASH (light blue). Without ASH state dynamics the chemotaxis index does not decay to 0. **c** Comparison of chemotaxis index for model with ASE (de) sensitization (red, orange) and with fully sensitized ASEL and ASER (black, brown). Source data underlying this figure are available in Supplementary Data 8.

sensitization and de-sensitization. Our simulations of naïve animals with and without ASE adaptation yielded similar chemotaxis indices in the quadrant assay over 10 min (Fig. 7e) and over an hour (Fig. 8c, black and brown lines).

In summary, our model predicts that dynamic-state switching of ASH mediates the behavioral switch associated with gustatory plasticity. Neither desensitization of ASEL nor sensitization of ASER appear to play a role in the quadrant assay, aside from their possible contribution to the recruitment of ASH.

**Sensory adaptation enhances exploration in an in silico salt spot assay.** The absence of an obvious role for ASE sensitization in the virtual quadrant assay raises questions about the possible benefit of ASE adaptation in *C. elegans*. We conjectured that ASEL and ASER adaptation may serve to modulate attractive search behaviors in more natural, heterogeneous environments. To test this, we constructed a virtual spot assay consisting of an infinite grid of identical spots with radial Gaussian concentration profiles (Fig. 9a). Model animals were placed in the center of one spot from which they were free to move up and down salt gradients, allowing them to visit different salt spots. Attraction to NaCl of the model animals was quantified as the fraction of time spent on NaCl spots (>25 mM), averaged over the population (Fig. 9b). Exploration behavior was quantified in terms of hops from one spot to another (Fig. 9c, d).

In this spot assay, model animals displayed a balance between localized attraction to NaCl and exploratory behavior (Supplementary Movie 15). Without ASH recruitment, adaptation-defective animals in which ASEL and ASER were fully sensitized (in Fig. 9) exhibited stronger NaCl attraction that led to the majority of animals remaining very close to their initial location for the duration of the simulation (Fig. 9b–d; Supplementary Fig. 8; Supplementary Movie 15–30). Incorporating ASEL and ASER (de)sensitization resulted in reduced local NaCl attraction and enhanced exploratory behavior (Fig. 9b–d; Supplementary Fig. 8). Enabling ASH recruitment further enhanced exploratory behavior (Fig. 9b–d; Supplementary Fig. 8). In our model, ASEL desensitization enhanced exploration during salt attractive behaviors by increasing the typical exploration radius of a spot (and hence the rate of escape from a given spot), ASER sensitization limited the attractive response to sufficiently large spots (or sufficiently long dwell-times on a spot), whereas ASH recruitment led to more widespread dispersal of the population (Fig. 9d; Supplementary Fig. 8).

Next, we determined the relative contributions of ASEL and ASER to exploration. We found that removing ASEL reduced exploration, whereas removing ASER in our virtual animals increased exploration (Fig. 9c, d; Supplementary Fig. 8). Unlike the quadrant assay, however, the contribution of ASEL to NaCl attraction was limited in the spot assay, whereas removing ASER had a stronger effect (Fig. 9b).

## Discussion

In naïve *C. elegans*, attractive, ASE mediated, and aversive, ASH mediated, salt responses are controlled by clearly delineated subcircuits, resulting in a switch between attraction up to 200 mM NaCl and avoidance of higher concentrations (Fig. 10). However, while naïve animals are attracted to low salt concentrations, extended NaCl exposure without food leads animals to avoid any NaCl concentration[2,5,6], implying an adaptive foraging behavior. Here, we showed that the behavioral switch between NaCl attraction and avoidance is mediated by plasticity in sensory neurons, resulting in altered dynamic ranges in both attractive and nociceptive subcircuits. Based on our experimental and

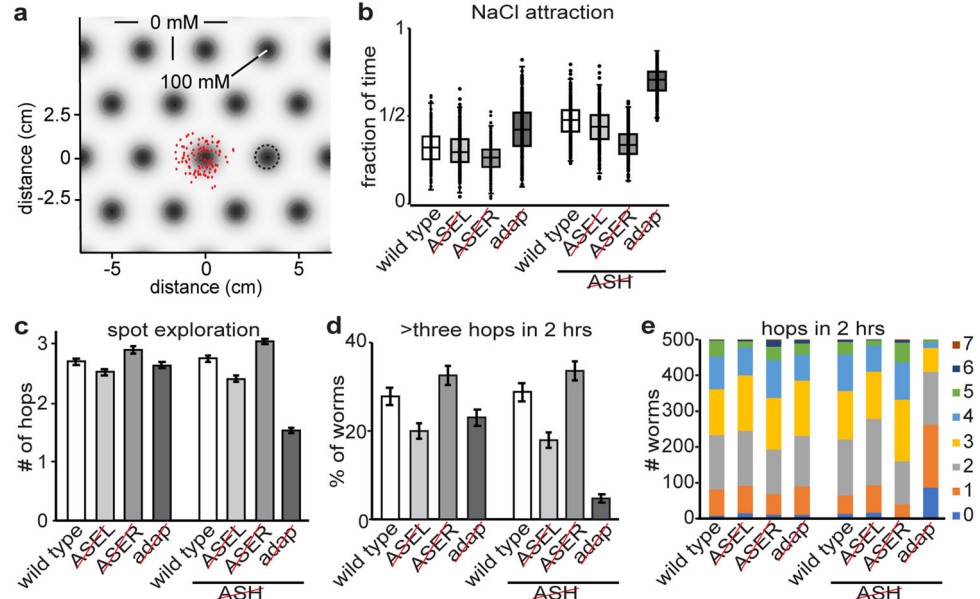

**Fig. 9 Sensory adaptation controls exploration in a virtual spot assay. a** Spot assay arena with identical salt spots with Gaussian salt concentration peaking at 100 mM. Peaks are positioned on an infinite hexagonal grid. The standard deviation for each spot is 0.6 cm and the distance between the two nearest spots is 3.3 cm. Populations of 500 worms each were simulated for 1 h. Naïve worms were initialized in the center of a single spot, with random orientations. **b** Salt attraction in the spot assay for eight different populations, measured as the fraction of time a worm dwells on salt (>25 mM).
**c–e** Exploration is quantified as the number of times a worm changes from one spot to another for eight different populations. For each population, 500 worm trajectories were analyzed. **c** The average number of hops achieved within 2 h (±SEM). **d** The percentage of worms that made four or more hops within 2 h (±SEM). Note the sixfold difference between the simulations with ("wild type", 28.8%) and without ("adap", 4.8%) desensitization in populations of model animals lacking ASH. **e** A breakdown of the number of animals that make 0–7 hops within 2 h. Source data underlying this figure are available in Supplementary Data 9.

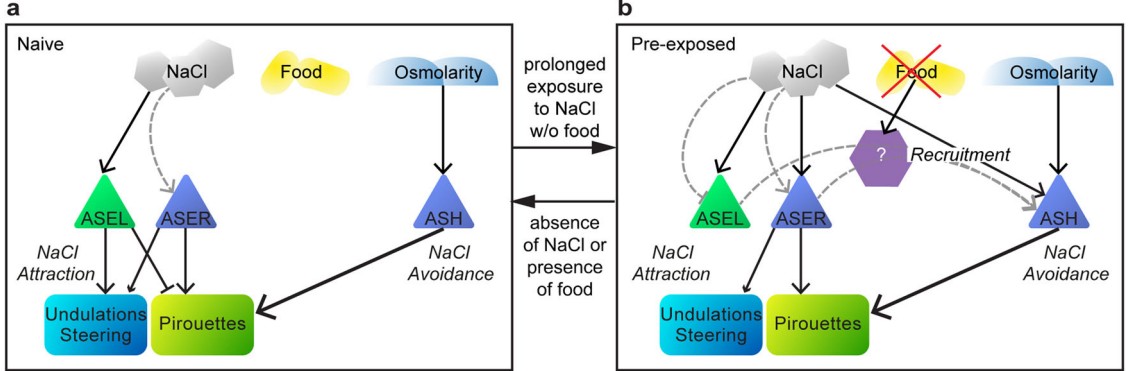

**Fig. 10 Schematic model of the NaCl navigation circuit.** Schematic of the different forms of sensory adaptation and their downstream effects in response to NaCl exposure in the absence of food. **a** The naïve state, in the absence of NaCl and/or the presence of food. ASEL is fully sensitized, ASER desensitized and ASH only responds to high NaCl concentrations (osmotic shock). **b** The pre-exposed state, after 10–15 min of exposure to NaCl in the absence of food: ASEL becomes desensitized, ASER sensitized and ASH recruited to respond to lower NaCl concentrations. Recruitment of ASH depends on an absence of food signal and ASE, possibly via one or more intermediate neurons. It is unclear whether an absence of food signals recruits ASH (as indicated schematically in the figure), or whether the presence of food inhibits the recruitment of ASH. ASEL and ASER mediate attraction to NaCl and ASH mediates avoidance of NaCl. NaCl-dependent adaptation is presented in gray dashed arrows. Solid arrows represent excitation (either via receptors or synapses), solid bars inhibition.

modeling results we propose that the sensory response of *C. elegans* to NaCl is regulated at multiple levels.

We observed an effective flip between the dynamic ranges of the primary avoidance neurons, ASH, and salt attractive neuron, ASEL, which suppresses attraction and enhances avoidance of naïvely attractive NaCl concentrations. Surprisingly, ASER, the second primary sensor mediating NaCl attraction sensitizes with NaCl exposure. Hints of opposite forms of adaptation in ASER and ASEL were already reported by Oda et al., who found adaptation after 10 min of exposure to 20 mM NaCl[25]. Our

analyses suggest that both ASEL and ASER (de)sensitization are mostly cell-autonomous. In contrast, sensitization of ASH requires signals from the ASE neurons, glutamatergic, serotonergic, dopaminergic and neuropeptide signaling, underscoring the complexity of this seemingly simple behavioral paradigm. One of these signals probably mediates the cue that food is lacking. Candidate cells for detecting the absence of food are the ASG pair of amphid sensory neurons, as these have been shown to play a similar role in taste avoidance learning[47]. However, it remains unclear whether the absence of food signals

recruit ASH and/or whether the presence of food inhibits the recruitment of ASH. Accordingly, our model remains agnostic to these possibilities, as the encoding of food in the model is implicit.

While multiple sensory neurons and contributions from the downstream circuitry most likely contribute to the rich behavioral responses in *C. elegans*, our simulations demonstrate the feasibility of a parsimonious model in which recruitment of ASH by an ASE (NaCl) derived signal underpins the switch between attraction and avoidance in gustatory plasticity.

The bilaterally asymmetric kinetics of the ASE neurons, in which ASEL and ASER depolarize in response to concentration increases and decreases, respectively[17], suggest a potential to double the ASE sensory dynamic range (from $\{0,x\}$ to $\{-x,x\}$), thus enhancing the resolution of NaCl sensing in the animal. However, the apparent timescale separation in the responses suggests otherwise. Our computational model demonstrates how the separation of timescales leads to distinct pathways that control different motor programs. ASEL and ASER have previously been linked to the control of steering and pirouettes, respectively[17]. In our model, fast sensory processing in ASEL controls steering, whereas slow sensory processing in ASER modulates pirouettes over tens of seconds or minutes. Such encoding of distinct motor actions in the kinetics of neuronal activation provides an effective mechanism for dictating behavioral output at any point along the sensory-motor pathway, even in the sensors themselves.

While one would expect some forms of adaptation in sensory neurons[3,48], our results point to severe information loss, causing a potentially considerable impediment in salt sensing. When ASER is desensitized the animal's ability to respond to concentration decreases is almost abolished. Our model predicts that naïve worms (with sensitized ASEL only) will move up gradients. If the NaCl regions traversed are insufficient to sensitize ASER, desensitization of ASEL will promote dispersion from NaCl-rich regions. Conversely, if ASER is sensitized, a trajectory down the NaCl gradient will be suppressed by promoting turning. Combined, ASE (de)sensitization would ensure that animals only respond to sufficiently large NaCl regions, ignoring small fluctuations. In summary, NaCl-adaptation of ASE neurons could serve to balance exploration and exploitation navigational strategies in complex, heterogeneous environments, as in our virtual spot assay.

In addition, if ASEL predominantly controls steering towards gradient peaks when navigating up the gradient, desensitization would reduce steering only after entering a salt region (promoting broader exploration within the region). Conversely, if ASER predominantly modulates pirouettes, then leaving a salt patch will likely induce a pirouette. The above reasoning is consistent with our model assumption that ASER mediates attraction only, an effect that is masked when ASH is recruited. In addition, ASER could mediate avoidance in gustatory plasticity by flipping its synaptic sign to a downstream interneuron in a food/starvation dependent way[49].

*C. elegans* feed on bacteria in patchy environments, most densely in rotting vegetation. These bacterial patches likely vary in size and may be well separated spatially, consistent with the *C. elegans* boom-and-bust life cycle[50]. As food is depleted, animals disperse and forage, often through highly variable and uncertain environments, which call for random foraging strategies. When near enough to a signal, active steering and turning are beneficial. Strategies involving the balance of exploration and exploitation make sense in this context. A previous study of food search, in an environment lacking any relevant signals, models the transition of *C. elegans* from local to global search as an optimal strategy for information gathering about the environment[51]. In that study, a minimal proposal for a tentative neural circuit combines sensory and modulatory neurons that feed into a decision-making unit downstream. In contrast, our study suggests that much of the adaptation is implemented in the sensory circuit itself.

The intuition presented here suggests that the compact nervous system of *C. elegans* may benefit from enhanced computation in sensory neurons at the price of considerable information loss. Taken together, our computational model and our and previous experimental data point to a highly complex set of distinct forms of plastic sensory computation in the NaCl sensing circuit, indicating that, compared to higher animals, *C. elegans* has seen a shift of computation from the inter- to sensory layers over its evolutionary history.

## Methods

**Strains and germline transformation.** The following strains were used in this study:

Wild-type *C. elegans* strain used was Bristol N2.
GJ243 *che-1(p679)I; gjEx513[sra-6::YC3.60 elt-2::GFP]*, 0x outcrossed
GJ254 *gjEx523[sra-6::YC3.60 elt-2::GFP]*
GJ282 *gpc-1(pk298)X; gjEx549[sra-6::YC3.60 elt-2::GFP]*, 6x outcrossed
GJ285 *gpc-1(pk298)X; gjEx552[sra-6::YC3.60 elt-2::GFP]*, 6x outcrossed
GJ1494 *odr-3(n1605)V; gjEx866[flp-6::YC3.60 elt-2::GFP]*, 6x outcrossed
GJ1497 *tph-1(mg208)II; gjEx866[flp-6::YC3.60 elt-2::GFP]*, 6x outcrossed
GJ1498 *tph-1(mg208)II; gjEx523[sra-6::YC3.60 elt-2::GFP]*, 6x outcrossed
GJ1499 *odr-3(n1605)V; gjEx523[sra-6::YC3.60 elt-2::GFP]*, 6x outcrossed
GJ2202 *gjEx866[flp-6::YC3.60 elt-2::GFP]*
GJ2208 *unc-13(e51)I; gjEx866[flp-6::YC3.60 elt-2::GFP]*, 1x outcrossed
GJ2209 *unc-13(e51)I; gjEx523[sra-6::YC3.60 elt-2::GFP]*, 1x outcrossed
GJ2210 *egl-3(ok979)V; gjEx866[flp-6::YC3.60 elt-2::GFP]*, 2x outcrossed
GJ2211 *egl-3(ok979)V; gjEx523[sra-6::YC3.60 elt-2::GFP]*, 2x outcrossed
GJ2213 *cat-2(tm2261)II; gjEx523[sra-6::YC3.60 elt-2::GFP]*, 1x outcrossed
GJ2214 *cat-2(tm2261)II; gjEx866[flp-6::YC3.60 elt-2::GFP]*, 1x outcrossed
GJ2218 *unc-31(e928)IV; gjEx523[sra-6::YC3.60 elt-2::GFP]*, 1x outcrossed
GJ2219 *unc-31(e928)IV; gjEx866[flp-6::YC3.60 elt-2::GFP]*, 1x outcrossed
GJ2223 *eat-4(ad819)III; gjEx866[flp-6::YC3.60 elt-2::GFP]*, 3x outcrossed
GJ2224 *eat-4(ad819)III; gjEx523[sra-6::YC3.60 elt-2::GFP]*, 3x outcrossed
GJ2277 *otIs204[ceh-32p::lsy-6 elt-2::GFP]; gjEx866[flp-6::YC3.60 elt-2::GFP]I, 1x outcrossed*
OH8585 *otIs4 [gcy-7p::gfp]; otEx3822 [ceh-36p::CZ-caspase3(p17) gcy-7p::caspase3(p12)-NZ myo-3p::mCherry]* (ref. 24)
OH8593 *ntIs1 [gcy-5p::GFP lin-15(+)]; otEx3830 [ceh-36p::CZ-caspase3(p17) gcy-5p::caspase3(p12)-NZ myo-3p::mCherry]* (ref. 24)

Germline transformation was performed as described, using an *elt-2p::GFP* construct as co-injection marker[52]. Promoters used for expressing the Yellow Cameleon (YC3.60)[21,22] construct were *sra-6* for ASH and *flp-6* for ASE.

**Cameleon imaging.** Images were acquired with a Zeiss Axiovert 200 M microscope, fitted with a Harvard apparatus MC-27 flow chamber. The naïve wash buffer contained 5 mM $K_2HPO_4/KH_2PO_4$, pH 6.6, 1 mM $MgSO_4$, 1 mM $CaCl_2$, the pre-exposure and stimulus buffers contained additional NaCl. The osmolarity of these buffers was set to 325 mosmol, using glycerol, except when NaCl concentrations were too high. Animals were glued onto 2% agarose pads using Nexaband® veterinary glue (World Precision Instruments, Sarasota, Florida). Stimuli were applied by moving a capillary into the buffer close to the nose of the worm. We used a custom automation in Improvision Openlab to control the movement of the capillary and to acquire the images. The acquired image was split into a CFP and YFP part with an Optical Insights Dualview beamsplitter (dichroic mirror 505 nm, 465/30 nm and 535/30 nm emission filters), and the intensities of the CFP and YFP fluorescent areas were recorded, normalized to the 2 s prior to the stimulus. The fluorescent ratio was determined by (YFP intensity)/(CFP intensity) − 0.6, where the 0.6 factor corrects the bleedthrough of CFP into the YFP channel.

**Behavioral experiments.** The response to 25 mM NaCl, with or without pre-exposure to 100 mM NaCl, was assessed as described before[6,42]. Briefly, animals were synchronized by bleaching and grown for 66–72 h at 25 °C. The animals were washed for 15 min with CTX buffer ($K_2HPO_4/KH_2PO_4$, pH 6.6, 1 mM $MgSO_4$, 1 mM $CaCl_2$) with or without 100 mM NaCl and (in a minimal volume) ~100 animals were transferred to the center of a quadrant chemotaxis assay plate (Falcon X plate). Two quadrants of the assay plate contained CTX agar (1.7% bacto agar, CTX buffer) with 25 mM NaCl, 2 quadrants contained CTX agar without NaCl; 15 min before the assay the quadrants were connected by a thin layer of CTX agar. Assay duration was 10 min except in the experiments presented in Fig. 8, where animals were followed for 60 min. A chemotaxis index was calculated: $(A − C)/(A + C)$, where $A$ is the number of animals at the quadrants with NaCl, and $C$ is the

number of animals at the quadrants without attractant. Assays were performed in duplicate, at least on two different days. The behavior of animals was always compared with controls performed on the same day(s). Animals that did not move away from the center or were located above the plastic edges were censored.

**Statistics and reproducibility**. All experimental results are given as a mean ± standard error of the mean (SEM). Statistical significance was determined using an ANOVA, followed by a Bonferroni post hoc test.

**Computational modeling**. Virtual worms were simulated in the quadrant assay[6,39] (Supplementary Fig. 13 in Supplementary methods) and in the spots assay (Fig. 9, Supplementary Fig. 14 in Supplementary methods). In the quadrant assay, every data point was run with 1000 animals; in the spot assay, we used 500. Quadrant assay simulations were initialized with worms at the center of a plate with Cartesian quadrants of alternating salt concentration. The interface between quadrants was modeled smoothly with a peak concentration gradient of 100 M/m (see Supplementary methods). Simulated behavior was quantified by the chemotaxis index as in the behavioral assay. The spot assay consisted of a hexagonal grid, defined by the spot radius, peak concentration, and spot separation distance. Chemotaxis was quantified by hop frequencies.

Model worms consist of a point worm with three sensory neurons ASEL, ASER, and ASH, a single downstream interneuron controlling the pirouette rate, and a simplified steering circuit. ASEL threshold modulation is used to model desensitization. Multiplicative gain modulation in ASER is used to model sensitization. ASH recruitment was modeled as a dynamic switch with rates that depend on the history of the NaCl concentration. Sensitization and desensitization rates were fit to match gustatory plasticity rates from the $Ca^{2+}$ imaging results. All virtual assays were simulated in the absence of food.

Steering was implemented by a half-center oscillator circuit capable of generating undulations as well as steering the worm, consistent with behavioral data and neuronal circuit motifs[38,53]. Model parameters were set so the pattern generation was achieved endogenously, but the same model circuit would support alternative (proprioceptively driven) control mechanisms[3].

To confirm the validity of the steering model to the quadrant assay, we qualitatively compared the behavior in simulations and experiments. As the strongest steering was observed near the quadrant boundaries, we systematically simulated model animals approaching and crossing the boundary at different angles of attack (Supplementary Fig. 6). Simulating the locomotion of simplified model animals with only one, fully sensitized sensory neuron and no pirouettes consistently showed that only ASEL influences the direction of the model trajectories across the sharp quadrant boundary, whereas ASER fails to steer the model animal. In these simulations, the comparatively faster rise time of ASEL (on the time scale of an undulation) is required to effect steering while the decay rate of the rectified ASEL response is immaterial. Finally, in our model, the rectification of ASEL, which limits steering to motion up the NaCl gradient is required to avoid negative chemotaxis when heading down the gradient.

Instantaneous changes of bearing to a new random direction were used to mimic a pirouette[3]. The pirouette rate was set by a single neuron, whose activation was up- or down-regulated by incoming negative (aversive) or positive (attractive) sensory signals. The spot assay base pirouette rate was set to 2.1 turns/min[26], whereas in the quadrant assay, a lower base pirouette rate was used to match experimental trajectories of naïve animals which predominantly used steering to orient themselves (0.3 turns/min in the quadrant assay, $n = 14$ animals).

Sensitization of ASER and ASH can lead to a competition between attractive and aversive responses to salt. To impose the aversive response, we set a much stronger ASH weight onto the pirouette interneuron than that from ASER, thus 'drowning' the attractive drive. An alternative "blocking" mechanism whereby ASH actively disrupts signaling along the ASE sensorimotor pathway is equally tenable. In fact, Oda et al. showed a complete loss of activity to NaCl downsteps in the AIB interneurons (postsynaptic to ASER) after pre-exposure to NaCl in the absence of food[25].

To validate our models, we confirmed that model animals exhibited similar motor behavior to those of animals in the quadrant assay (Supplementary Fig. 12 in Supplementary methods). Further modeling details are given in Supplementary methods.

**Reporting summary**. Further information on research design is available in the Nature Research Reporting Summary linked to this article.

## Data availability
Data underlying the main figures are presented in Supplementary Data 1–9. Other data generated or analyzed during this study are included in this published article (and its supplementary information files) or available from the corresponding author on reasonable request.

## Code availability
Code generated and used in this study is available with instructions at http://www.wormlab.eu/plasticity/.

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

## Acknowledgements

We thank R. Tsien, A. Miyawaki and B. Schafer for constructs, B. Schafer and H. Suzuki for help in setting up imaging and for sharing unpublished results, WormBase for providing annotated data on prior work, and the *Caenorhabditis* Genetics Center for strains. We also thank C. Brittin, R. Gudde, I. Hope, R. Hukema, S. Lockery, S. Rademakers and T. Thiele for suggestions. This work was funded by the Center for Biomedical Genetics, the Royal Netherlands Academy of Sciences, ALW/NWO and the EPSRC via grants EP/J004057/1 and EP/N010523/1.

## Author contributions

M.P.J.D., F.S., T.S., O.U., N.C., and G.J. designed the research and analyzed and interpreted the data. M.P.J.D., O.U., and G.J. performed *C. elegans* genetic, imaging, and behavioral experiments. F.S and T.S. developed computational models. M.P.J.D., F.S., T.S., N.C., and G.J. wrote the manuscript.

## Competing interests

The authors declare no competing interests.
