## [Peer Review File · Communications Biology]

Reviewers' comments:

Reviewer #1 (Remarks to the Author):

The manuscript "Plasticity in gustatory and nociceptive neurons controls decision making in *C. elegans* salt navigation" by Dekkers and colleagues is an interesting piece of work on *C. elegans* behavioral plasticity and salt learning, which combines experimental and mathematical efforts. The coupling of experiments and mathematical modeling is definitely a strength, as it enriches the tools that can shed light in *C. elegans* neurobiology. The intention to introduce the question of timescales and neuronal dynamics is much appreciated, as well as the goal to distinguish between different roles of neurons, depending on the environmental conditions and the recent past. At the same time, the manuscript presents certain weaknesses, which pertain mainly to how experimental (mostly) results are presented, contextualized and interpreted, how certain choices are justified, and how the findings are discussed in a broader context. The manuscript would also benefit by restructuring/revising certain parts. The discussion of findings needs to be updated, taking in to account a number of recent publications, relevant to the topic. More detailed comments are provided below.

General

1. The absence of Lines numbers makes things quite difficult when the need to refer to a particular phrase/paragraph arises. Due to this, I quote the parts I need to comment on.
2. The practice of discussing the results of >half the panels of Fig 1 after Fig 4 is discussed, is very confusing. Same for Fig. 3. Authors should strongly consider changing the order of the figures/panels if the current order does not serve the flow the wish to follow.
3. Some language polishing might be needed here and there.
4. Some claims might need to be clarified or restated in a more careful way. In some cases, additional data analysis might be needed (eg quantification of decay times)
5. The mathematical component results are written in a more precise and straightforward way than the experimental part, which makes the manuscript uneven, although the clarity of the math part is a strength.
6. The mechanism of neuronal dynamics is not explored in depth (ie rescue experiments); this is not necessarily an issue, but maybe some of the claims should be stated more carefully.

Abstract

1. What are the "high-level" behaviors the authors refer to? Decision making? Other? It should be clarified.
2. "Sensory adaptation inevitably results in less reliable encoding": the use of the term inevitably might be too much and a little unjustified. Why does adaptation inevitably lead to less reliable encoding??
3. "sensors encode the subjective value of the relevant stimulus": The term subjective might be confusing, since a reader might understand that it differs from one individual to another, but I guess this is not what the authors mean. My understanding is that they want to refer to the recent past of the animal or the context/environmental conditions. The claim here should be revised.
4. How does the computational model account quantitatively for the observed behaviors? What is the trait/feature that is quantitatively captured by the model?
5. Some mention of what is already known about *C. elegans* salt learning should be made, so that the reader could place this work in the appropriate context.

Introduction

1. The intro lacks focus and cohesiveness and fails to provide the reader with a fair idea of what are the premises of this work and consequently which is the niche the authors wish to fill. As a result, the reader is not well prepared of what there is to follow and is not provided with baseline information. Consider, for example, answering the question: what is the scope of this paper and why? Half of the intro is a summary of the findings, but it does not serve its role to introduce the reader to the work.
2. Adaptive decision making: the authors might wish to build a case for the term here, as it relates

to the dynamic aspect of things, which is partly the focus of the mathematical model.

3. Consider avoiding two antithetical sentences back to back in the first paragraph.
4. "decisions are generally thought to be the result of integration of information from sensory neurons in downstream interneurons": yes, and also given the animal's prior experience.
5. "In the absence of a reward, the expected value of an action will wane": the absence of a punishment has an analogous role in decision making.
6. "Like most animals, naïve *C. elegans* are attracted to NaCl": you need to provide a citation for the "most animals" part, which I am not sure is the case, so generally speaking.
7. It might be preferable to refer to [NaCl] or to NaCl concentrations instead of plain NaCl when authors write things like NaCl decrease or increase.
8. "ASEL is desensitized upon pre-exposure to NaCl; ASER is initially insensitive to NaCl changes and sensitizes upon pre-exposure; ASH nociceptive neurons become sensitized to considerably lower (non-toxic) levels of NaCl upon pre-exposure.": I understand this summarizes many of the findings; however, it reads kind of confusing. Authors might want to avoid comparative terms (initially, lower: initially, meaning before what? Lower than what?) and restate the findings in a clearer and more enticing way.
9. The Weber law needs to be explained here. Its relevance to the findings or the hypothesis needs to be provided, too.
10. "We show that a hierarchy of molecular, cellular and distributed circuit mechanisms are necessary and sufficient to fully capture our Ca²⁺ imaging results": necessary and sufficient is a string statement. Authors need to reconnect with it and justify this claim later in the Discussion in a convincing way, if they wish to leave the claim here.

Results

1. I am not sure I follow the authors' logic of which figures to present as main figures and which to include in the Suppl Information section. The opening part of the Results describes findings showcased in a Suppl Figure, which is weird, and it does not help the reader understand where is the focus of the paper, which information is of primary importance and which is accompanying information.
2. "Naïve sensory responses": it could also include that these are short term exposure responses, like the caption of Fig S1, see also headline of the next section.
3. "but only rarely to lower concentrations": what does "rarely" mean?
4. "led us to conjecture that ASER responses": so is this a hypothesis? Do the authors hypothesize that? If so, then they can continue by saying "In order to test this hypothesis..." The manuscript could benefit significantly from stating the course of action in a hypothesis-driven way.
5. "We found that the propensity to respond, as well as the amplitude of the response increased with exposure time": the propensity to respond is hard to quantify, how do the authors achieve that? Please clarify.
6. Supplementary Table 1: Nice idea to provide all this information in one place. However, the table is not as helpful as it could be. Please provide the Fig number next to the reported response/neuron/experiment. Provide these data for all panels of all figures. I suppose there are "censored" animals, ie the ones that did not respond. Could this constitute some kind of "response index"? Could this be important? Does a "response index" or percentage regarding Ca transients correspond to the indices of behavioral assays? Importantly: Why is the rise time (mean and range) missing for some of the responses? ASEL with pre-exposure: why are so few animals tested here, compared to the rest of the experiments, eg ASEL no pre-exposure? In the caption, it reads "ASER stimulus steps are given in pre-exposure time": this can be confusing. Please insert new columns/rows if needed, but please provide the info in a more straightforward way. This table is the identity of the experiments, so it needs to be well written. Regarding the ASH experiments (I did not try that for the rest): Exposure to 200mM [NaCl], no preexposure 7/18 animals respond 39%; preexposure 600sec 16/17 94% animals respond. Similarly, for 300mM: 68% vs 90% of animals; for 500mM the %s are similar. How do the authors comment on this? Is this important in their opinion?
7. Suppl Table 1, Fig 1, and in general: I think that a computation of the rising slope would help a lot with data analysis. Even more than rising time, mean and range, does. The slope can be very informative of the dynamic behavior of the system (neuron). Moreover, it might make a difference if used instead of the rise time to the max amplitude, like the authors do below (I have a comment on that later). The slope reflects how fast the Ca channels respond on the neuron's membrane.

8. As mentioned before, the practice of discussing the results of >half the panels of Fig 1 after Fig 4 is discussed, is very confusing.
9. Fig 1: Is there a graph to show how wild type ASER responds to exposure following pre-exposure? Like Figs 1c, d, e?
10. What is the n for 1d and 1e?
11. "To determine whether ASEL responses are also modulated by pre-exposure, we exposed animals to 100 mM NaCl for 60 seconds to 10 minutes and, after a 30 - 60 seconds wash, tested their response to 100 mM NaCl": I think 30 sec wash is not shown in Fig or in Table.
12. "Hints of opposite forms of adaptation in ASER and ASEL were already reported by Oda et al., who found weak adaptation after 10 minutes of exposure to 20 mM NaCl¹⁹": I think this might be better off in the Discussion.
13. Supplementary Fig. 3: How many animals tested (n)? What is the standard deviation or SEM? Is the chemotaxis index obtained in 50mM significantly different than the one for 10mM?
14. Fig. 2: Have the authors conducted a time course study for ASEL, like they did for ASER? Why not 30 sec exposure for ASEL, to compare with ASER?
15. "We conclude that the ASEL response to NaCl desensitizes with pre-exposure and recovers in the absence of NaCl exposure, and conjecture that this sensory adaptation modulates the strength of attraction to NaCl in behavioral assays.": First of all, I think that it is the neuron that sensitizes or desensitizes, not the response. Second, maybe this passage could fit better in the Discussion section. Instead of conjecture, would "suggest" be a good alternative here?
16. "We found response rates, amplitudes and profiles comparable to naïve responses to 500 mM NaCl": Have the authors conducted some statistics to support that? How are these features, ie "profiles" comparable? "Thus, ASH neurons are sensitized by pre-exposure to NaCl.": This too general a statement, maybe the authors would like make it more precise, so that it matches their findings.
17. "is prevalent in many nervous systems, including C. elegans²¹": C. elegans is not a nervous system, please be careful with wording.
18. Authors should provide a justification for i) the neurons they are interested in, and ii) the mutants they have selected. Citations are good, but a proper justification needs to be provided some place, preferably the Results (alternatively the Discussion).
19. Why is Suppl Fig 4 a Supplementary figure? Why do authors provide only part of the Ca transient in Suppl Fig 4a (odr-3), c (egl-3)? What happens for the rest of the 600 sec?
20. "five out of nine animals tested showed a weak response to 100 mM NaCl after 10 minutes preexposure": can the authors provide that data? It might be important for the rest of the argument.
21. "Culturing tph-1 animals for 72 hours on plates containing 2 mM serotonin resulted in complete desensitization of ASEL in 7 out of 9 tph-1 animals tested": this is very interesting trend, but the data is few (compared to 5 out of 9 of the other case), the sample size is small; if 1 or 2 worms had behaved differently, the final conclusion would different, but the sample size is small. I suggest the authors do not rely on these data this much, can the difference be evaluated by a statistical test? In relation to this, "hinting at an additional serotonin-dependent desensitization mechanism.": hinting is ok, but I think the relation to serotonin should not be based on the serotonin enriched plates data (see above).
22. Moreover, the whole story of cell-autonomous response, I think the data really alludes to that, but it would take more mutants or even neuronal ablation of upstream neurons to make such a claim. Maybe the authors can re-state the claim? Besides, the serotonin data might be actually pointing the other way, if we had more data points we could tell with less uncertainty.
23. Related to that: are the authors certain that the strains they got have been outcrossed? And outcrossed sufficient times? Maybe the different responses among worms of the same strain are related to background mutations, very common if the strains are not backcrossed. For example, odr-3(n1605) is not outcrossed (see wormbase); unc-13(e51) I is not outcrossed, either. (Related to that, the authors should provide the strains names in the Methods section).
24. Fig. 3, what is the n for all the panels?
25. "Transient pulse-like responses are well captured by two opposing and timescale separated components." Please provide citations of other papers that report similar findings.
26. "are gap-junctionally coupled": unusual selection of adjective instead of a noun here.
27. "Hyperosmotic responses were not considered": I guess for simplicity, but the authors need to

justify that.

28. Fig 4: please consider changing the combination of black font on blue background wherever applicable.

29. "We parameterized the model using Ca²⁺ imaging data from fully sensitized ASEL and ASER neurons. Both the rise and decay times of the responses were consistently and significantly faster in ASEL than in ASER, consistent with previous work⁶": this is interesting, because nowhere in the experimental results is the decay time quantified/calculated. I suggest the authors do that, otherwise the comparison is not supported. "the ASER decay time is much slower than that of ASEL.": again, I could find no data for this. Moreover, why is this repeated twice? Do the authors need to emphasize on something? Importantly: the amplitude is different in ASEL and ASER responses, indeed. Until ASEL max, though, the slope seems to be the same (they completely overlap in Suppl Fig 6). Could this mean that the dynamics/kinetics of the channels are the same, only the max amplitude differs? This might be important. Relevant to that: is the initial baseline similar/different?

30. Fig 4 is not a results figure; it is more like a methods/approach figure. Should another figure be used to report the modelling results? Maybe fig M3?

31: "consistent with the absence of a response in the naïve context": where is this shown?

32. Fig. 5: what does the strikethrough indicate?

"(a) Our model shows no difference in attraction in the quadrant assay between wild type and animals with always fully sensitized ASEL and ASER": have the results of these animals been pooled? Why are there two bars in the graph?

Moreover, "wild type" is usually juxtaposed to mutants. There is confusion when wild type is juxtaposed to (de)sensitized, as I think is the case here, since these worms are "genetically" wild type. Maybe pick a different term?

"(b) Ablating the synaptic connection from ASEL to the steering motor neuron or from ASER to the pirouette neuron significantly reduced the chemotaxis index.": which model parameters correspond to that?

"(c) Double mutants with ablated ASEL and no steering, or ablated ASER and no pirouettes": double mutants is a weird term for model animals, maybe the use of quotes would help here.

"(d) Experimental validation: animals with genetically ablated ASEL (OH8585) and genetically ablated ASER (OH8593) exhibit a strong reduction in the chemotaxis index in the quadrant assay": the authors here name the strains, which they do not do (although they should) for the rest of the strains in Methods. Importantly: since the authors have these ablated strains, have they tracked them using a worm tracker, to see whether their locomotion (ie pirouetting, steering) differs from wild type (intact) animals? I think this would help support their claims and validate further their model.

33. "A sensitivity analysis on the key sensory parameters (ASEL/R rise and decay times) showed that, across the entire parameter space, introducing ASE adaptation led to significantly lower or at best equal chemotaxis (data not shown).": actually, a sensitivity analysis would be very interesting to see. Plus, what if you tweak the sensitization parameters or the [NaCl] value?

34. In ASEL and ASER ablated animals, how do the Ca transients change, in ASH and in non-ablated ASE neurons?

35. "Without any further parameter tuning, simulations of naïve and pre-exposed animals closely reproduced these experimental data": this is a strength and should be highlighted.

36. First paragraph of page 17: I think there is repetition here, and in addition maybe something is missing.

37. I am not sure I understand the stochastic recruitment of ASH. Does this happen randomly? Can the authors elaborate on this, maybe in the discussion?

38. Fig. 7: this is indeed a very interesting in silico experiment. I think it makes sense that they are ~1cm, ie ~1 worm body length, this is something I can see at the suppl video. In 7c, the change does not seem to be that much in all cases, similarly in 7d, although here a 30% change sounds more convincing. How big an area does this correspond to? How is this different than a no spot-containing area? Maybe this would help authors illustrate better their claim. In addition, what is the max number of hops (to add to 7d)? this could also help highlight any differences. Or, how fast do they move from one hop to the next?

By watching the Suppl Videos 1-4, I was also wondering the following: How about the fraction of worms that leave the initial spot? And how about overall dispersal? In ASH-ASE (de)sens, are there secondary clusters formed or is it just mu idea? Lastly, in -ASH video, I think they move

faster, can the speed be quantified? Also, what about the total area covered?

39. How do the authors discuss their findings in the context of Klein et al, 2017 (doi: 10.7554/eLife.30503, Samuel and Karplus groups) and Calhoun et al, 2014 (doi.org/10.7554/eLife.04220.001, Chalasani and Sharpee groups)? No salt chemotaxis here, but exploration behavior.

Discussion

1. Fig 8 and discussion in general: I am confused about the absence /presence of food. Do authors incorporate data from the literature in the schematic of Fig 8? Because as far as I can understand, the presented experiments do not explore both conditions.

2. In relation to that: "our simulations demonstrate the feasibility of a parsimonious model in which recruitment of ASH by a food and an ASE (NaCl) derived signal underpin the switch between attraction and avoidance in gustatory plasticity": where is the food signal in the model?

3. "C. elegans has seen a shift of computation from the inter- to sensory layers over its evolutionary history": this is mentioned in Iino, 2013, Salt Chemotaxis Learning in *Caenorhabditis elegans*, doi.org/10.1016/B978-0-12-415823-8.00013-7, Handbook of Neuroscience: "learning is in the sensory neurons". How is this claim novel here?

4. In general: I appreciate authors' effort to keep discussion short, nevertheless I would like to see the results discussed more extensively in the context of similar findings and of *C. elegans* ecology, evolution and environmental needs. A short paragraph with basic conclusions/contributions to close the Discussion would also help.

5. How do the authors discuss their results in light of work done by Liu et al, 2018 (doi: 10.7554/eLife.36419, Leifer Lab) on context dependency of *C. elegans* response to mechanosensation?

How do the authors discuss their results in light of the paper by Jang et al, 2019 (<https://doi.org/10.1073/pnas.1821716116>), Iino Group, on starvation-dependent aversive navigation in *Caenorhabditis elegans*, which includes results on ASER?

Methods

1. Where animals synchronized? What age were the worms tested? "Adult" is a little vague.

2. What does "if possible" mean, regarding the use of glycerol to regulate osmolarity?

In behavioral experiments: how many worms per plate? How many repetitions? Any censored animals?

3. "Assay duration ranged from 10 to 60 min but was 10 min unless specified otherwise": can the authors explain the range in mins?

4. "Model parameters were set so the pattern generation was achieved endogenously, but the same model circuit would support alternative (proprioceptively driven) control mechanisms²¹": why is this mentioned here? To show versatility of the model?

5. Have the authors considered trying a simulation with ASEL and ASH?

6. Could the authors provide a figure with experimental trajectories (quadrant assay) and simulation trajectories? It is mentioned that adjustments were made so that simulation trajectories match experimental ones, so it would be useful to see a few.

Supplementary Methods

Computational Method

1. I think a schematic of the model circuit would help.

2. Figure M1: is the DMN to DM connection really inhibitory?

3. Can the authors elaborate (last paragraph of page 12) on how Ca transients data has shaped modeled sensory neurons?

4. Tables with parameters values: are all values determined in this work or are some based on values from the literature?

5. Fig. M3b: is there an experimental current available from ASEL, to superimpose, as is the case for the other two neurons?

6. The Fechner's law appears to be quite central to this work, as it appears in various (at least 3) places in the manuscript. Would the authors consider devoting a paragraph somewhere to state the law and expand a little on its implications for organisms, providing a few examples? I think it would help with the overall narrative.

7. ASH recruitment: are there any examples in the literature of other neurons that behave

similarly? Or other similar cases, more general?

8. Parameter κ is the common rate parameter for recruitment kinetics; can the authors provide a citation?

9. Suppl Video S5: is this peak concentration 100mM? Then why is the color bar set with max at 200 (blue) and the spots colored blue? Similarly, is Suppl Video S9 with peak concentration 200mM? Then why is the color bar the same with S1 (max 100mM)? S13 seems ok, on the other hand.

Also, I think in S5 the dispersion distance between spots is not 5.0cm (compare to S13, for example). Is there any possibility that the videos have been mixed up? Please double check the sets of videos.

10. Is it possible to provide simulation videos of the simulated quadrant assay?

Reviewer #2 (Remarks to the Author):

Summary:

Dekkers and Salfeder et al., conduct a thorough analysis of the neuronal activity and adaptation of the main NaCl sensory neurons ASEL, ASER and the nociceptive ASH neurons. These analyses extend previous reports on de-sensitization of the ASEL and sensitization of the ASER neuron following exposure to low NaCl, and reveal that ASH also exhibit sensitization to lower concentrations. The authors conduct a genetic screen that suggests that (de)sensitization of ASE neurons is cell-autonomous, while the sensitization of ASH neurons requires the ASE neurons and various neurotransmitters. Lastly, the authors use the information regarding the NaCl sensory circuit to devise a simplified model of circuits and behavior to generate a comprehensive simulation. The model uses the NaCl sensory circuit response and adaptation to guide NaCl chemotaxis behavior. The simulation suggests that time-scale differences between ASER and ASEL guide pirouettes and steering behaviors, respectively, and avoidance after salt exposure mainly mediated by ASH. Efficient NaCl chemotaxis in the simulated quadrant assay did not require gustatory plasticity, at initial time-point but an experimentally observed temporal decline in chemotaxis index did. Simulating the behavior in a more complex "spot" assay suggests that plasticity in ASE and ASH neurons plays an important role in balancing aversion and attraction to achieve an optimal balance between exploration and chemotaxis behavior in more complex environments. I think these are interesting data and modelling results that provide a consistent model about gustatory plasticity in *C. elegans*. I see a conceptual advance in showing that gustatory plasticity at least in part could be explained by changes in a network of sensory neurons. I would be very enthusiastic about publication if the concerns below can be addressed.

Major points:

1) Does the simulation of the ASE neurons recapitulate their activity dynamics?

See Supplementary Figure M3.

In this figure the simulated ASE and ASH neuron activity is plotted. However, the activity of the experimental values of ASEL and ASER neurons is plotted only for one, fully sensitized condition (for example for the ASEL the 30s pre-exposure to 100mM NaCl followed by a 30 seconds wash). While the model was parameterized according to the fully sensitized Ca²⁺ experimental data, it is important to show it can re-capitulate the (de) sensitized states of these neurons.

To my understanding the relevant experimental data for ASEL exposure to 60,120,300 and 600 seconds of 100mM NaCl is present in Figure 2a.

Same for the different wash times, the simulated data is presented in Figure M3b, while the experimental data in Figure 2e. They should be also shown side-by-side.

From my observation they do not seem to fully match the simulated responses, this should be addressed.

2) One key point observation is the different effects of ASER and ASEL on behavior relate to the different time constants of their Ca²⁺ response. For example It is claimed that "... a significant timescale separation between the Ca²⁺ responses of fully sensitized ASEL and ASER, both in their rise and decay times (Supplementary Fig. 6)." and also regarding the consequences on the model performance in simulated behavioral assays " Therefore, to effectively modulate this rate requires a memory of salt exposure over commensurate (or longer) timescales. The fast decay time of ASEL precludes this, while the slow decay time of ASER is ideally suited to modulate the pirouette rate effectively. "

This is based on the result presented in supplementary figure 6.

And a result from a previous study (H Suzuki and TR Thiele et al., et al Neuron 2008, Figure 1).

While the slower decay response of the ASER neuron is convincing, I am not convinced that the rise of the ASER neuron is faster as claimed.

As this plays an important role in the model and its interpretation, I think it would be beneficial to show how the results of the model change when the rise and/or decay rates of ASER and ASEL are equalized.

I am also curious to know how the authors view the different roles of ASER and ASEL in regulating runs and pirouettes with respect to the following previous publication (H Suzuki and TR Thiele et al., et al Neuron 2008) that "activation of ASEL elevated forward probability (Fig. 3c, ANOVA, $p < 0.01$) whereas activation of ASER had the opposite effect (Fig. 3d, ANOVA, $p < 0.05$). We conclude that ASEL activation causes runs whereas ASER activation causes turns; thus the functional asymmetry between ASE neurons extends to the level of behavioural output."

Moreover, consistent with both models, the effect of genetic ablation of ASEL and ASER shows they are both required for NaCl chemotaxis. Analysis of pirouette and steering behaviors in these backgrounds during chemotaxis would allow the authors to validate their results and conclusions from the model, with respect to selective of ASEL and ASER in modulating these two behaviors.

3) I think the following supporting data should be shown; it is important part of the logic of the paper.

"A sensitivity analysis on the key sensory parameters (ASEL/R rise and decay times) showed that across the entire parameter space introducing ASE adaptation led to significantly lower or at best equal chemotaxis (data not shown). Therefore, a model that includes sensory adaptation is constrained to the small region of parameter space in which the detrimental effects of sensory adaptation are minimal if this model result reflects a true ... it begs the question what advantage is gained by sensory adaptation"

5) Regarding the statements on the role of serotonin in ASEL adaptation -

"Only tph-1 mutants showed slightly abnormal ASEL adaptation, suggesting some..."

and later

"while hinting at an additional serotonin-dependent desensitization mechanism"

-The effect side shown for the tph-1 mutants and serotonin Supplementary Fig 4c is very small and therefore I believe these statements should be moderated.

6) Regarding the finding in Figure 6b about the important role of ASH in plasticity induced avoidance, in comparison to the smaller role of ASE - validating these results with genetically or laser- ablated ASH animals, is needed to support their claims. Although, the authors refer to an earlier publication (Hakema 2006) on page 14, in Figure 3B of this publication gustatory plasticity defective in odr-3 is not rescued by odr-3 expression in ASH.

7) Page 17 Figure 7d and supplementary figure 9

"Enabling ASH recruitment further enhanced exploratory behavior (Figure 7b-d, supplementary fig

9.)."

Figure 7b-d does not show higher exploratory behavior (less or very similar # of hops and % of worms with three hops – comparing WT to ablated ASH WT).

In contrast supplementary figure 9 shows higher exploratory for ablated ASH.

The authors should explain these different results in the text, and in general clarify this part.

8) The authors used a stochastic switch model for ASH recruitment. The rationale for this model design is unclear and not explained at all. Inspecting the numbers of non-responders vs responders in Suppl. table 1 does not justify stochastic recruitment of ASH versus ASE. Is this a plausible model? More explanations are needed here in results text and discussion.

9) Throughout the paper, statistics are performed only for selective sets of experiments. For example, data shown in Figure 1c-e lack any statistics. This needs to be fixed throughout all experimental results.

10) Comparing the n numbers in Suppl Fig 1a-b, it seems only responders were used here for quantification and plotting. What is the justification for this?

Minor points –

-(Optional) in discussion "To our knowledge, this would be the first example of sensory neurons encoding and controlling distinct motor actions" I am not sure that statement is correct.

- Controls should be shown in the relevant figures, not in a separate figure to allow comparison. This is relevant to the following figures:

-Figure 3 - a control of ASH response to 200mM NaCl in naive worms not pre-exposed to NaCl is missing to allow comparison (like the one shown in supplementary figure 1d).

-Supplementary figure 4 - the ASEL desensitization response in wild type (like the one in Figure 2c) should be shown again in this figure to allow a comparison.

-Page 9 under "Sensitization of ASER is also predominantly cell-autonomous"

"we found a strong response of the ASER neuron of unc-13(e51) animals.. (Fig 1c.)"

Is it ok to refer to figure 1c only so late at the paper after figure 3 has already been presented?

- The phrasing of the legends of the figures is a bit messy, and the sentences can be clarified better. For example: legend of Figure 1 (Page 5) "did not result in response, 10 minute of exposure did albeit 2 seconds later than in wild type" - this is a very lengthy sentence that is not very clear.

- The reference to the previous publication of Oda et al (J Neurophysiol 2011) "Hints of opposite forms of adaptation in ASER and ASEL were already reported by Oda et al., who found weak adaptation after 10 minutes of exposure to 20mM NaCl. "

The adaptation in Oda et al paper (Figure 1) is not weak

Other authors recognized that the effect is there citing Oda et al

For example Katz et al., Cell reports 2018

"Imaging of ASER reveals that salt cues usually induce calcium signals that drive synaptic vesicle release. Following starvation, however, neuronal outputs are inhibited even though calcium signals are enhanced (Oda et al, 2011). "

Beets et al., Worm, 2013

". This is supported by the observed change in neuronal activity of ASE neurons following the short pre-exposure of worms to salt in the absence of food 55 . "

Page 7 under "Prolonged exposure to NaCl sensitizes ASH" - a not clear phrasing of sentence

"We found response rates, amplitudes and profiles comparable to naive responses to 500mM"

This phrasing is a bit confusing. The important comparison here is not pre-exposed ASH versus

pre-exposed ASH, when exposed to the same NaCl concentration of 200mM. The ASH neuron in naive worms is not responding while ASH in pre-exposed worms is responding. Therefore, I think it is better to compare the results of the sensitized ASH to the non-sensitized ASH and only then note that these responses were like the responses to high concentrations.

-Figure 7b-d – since average of 1000 simulated worms is presented, it would be more informative to show violin plot or another representation that shows how the distribution of the values (perhaps there's a bi-modal distribution?)

-Figure 6b. why ASH 100mM and 0mM ASH-on are not presented?

Plasticity in gustatory and nociceptive neurons controls decision making in *C. elegans* salt navigation

Reply to the reviewers

We thank the reviewers for their very detailed comments on our manuscript. We have revised our manuscript accordingly. Please find our response to the reviewers' comments below.

Reviewer #1

*The manuscript "Plasticity in gustatory and nociceptive neurons controls decision making in *C. elegans* salt navigation" by Dekkers and colleagues is an interesting piece of work on *C. elegans* behavioral plasticity and salt learning, which combines experimental and mathematical efforts. The coupling of experiments and mathematical modeling is definitely a strength, as it enriches the tools that can shed light in *C. elegans* neurobiology. The intention to introduce the question of timescales and neuronal dynamics is much appreciated, as well as the goal to distinguish between different roles of neurons, depending on the environmental conditions and the recent past. At the same time, the manuscript presents certain weaknesses, which pertain mainly to how experimental (mostly) results are presented, contextualized and interpreted, how certain choices are justified, and how the findings are discussed in a broader context. The manuscript would also benefit by restructuring/revising certain parts. The discussion of findings needs to be updated, taking in to account a number of recent publications, relevant to the topic. More detailed comments are provided below.*

General

1. The absence of Lines numbers makes things quite difficult when the need to refer to a particular phrase/paragraph arises. Due to this, I quote the parts I need to comment on.

Reply: Apologies for not using line numbers. That would indeed have been very handy. We have now included them.

2. The practice of discussing the results of >half the panels of Fig 1 after Fig 4 is discussed, is very confusing. Same for Fig. 3. Authors should strongly consider changing the order of the figures/panels if the current order does not serve the flow the wish to follow.

Reply: We agree with the reviewer that it is clearer to first show the naïve responses in a figure and then show the responses of the different cells. In the original submission we chose to present the data per cell and show the naïve responses in the supplementary data, to reduce the number of figures. We have now adapted the figures as suggested by the reviewer and have also rearranged other figures so that we still have 10 figures.

3. Some language polishing might be needed here and there.

Reply: We have edited the manuscript extensively. We have considered submitting a manuscript with all changes tracked, but this resulted in an unreadable manuscript. Important changes have been discussed at the points raised below.

4. Some claims might need to be clarified or restated in a more careful way. In some cases, additional data analysis might be needed (eg quantification of decay times)

Reply: We have clarified or restated our claims as suggested by the reviewer. Decay times are now given in lines 379-381.

5. The mathematical component results are written in a more precise and straightforward way than the experimental part, which makes the manuscript uneven, although the clarity of the math part is a strength.

Reply: We have substantially rewritten the experimental part to make it more precise and straightforward.

6. *The mechanism of neuronal dynamics is not explored in depth (ie rescue experiments); this is not necessarily an issue, but maybe some of the claims should be stated more carefully.*

Reply: We have gone through our claims and where possible referred to previously published data that support them. Where necessary, we have rephrased our claims and indicated confirmation would be necessary, e.g. by doing rescue experiments or using a second mutant allele (see lines 250-251 and 274).

Abstract

1. *What are the “high-level” behaviors the authors refer to? Decision making? Other? It should be clarified.*

Reply: We have changed “high level behaviors” to “motor behaviors, decision making and navigational strategy”.

2. *“Sensory adaptation inevitably results in less reliable encoding”: the use of the term inevitably might be too much and a little unjustified. Why does adaptation inevitably lead to less reliable encoding??*

Reply: The term “inevitably” indeed presupposed too narrow an interpretation of stimulus encoding. We have reworded the relevant sentence to be more precise.

3. *“sensors encode the subjective value of the relevant stimulus”: The term subjective might be confusing, since a reader might understand that it differs from one individual to another, but I guess this is not what the authors mean. My understanding is that they want to refer to the recent past of the animal or the context/environmental conditions. The claim here should be revised.*

Reply: We have rephrased this paragraph to clarify the context of this statement. The term subjective has been removed from the abstract.

4. *How does the computational model account quantitatively for the observed behaviors? What is the trait/feature that is quantitatively captured by the model?*

Reply: The outcome of the quadrant assay, the chemotaxis index (CI), is quantitatively captured by the model. We use the CI as a metric for evaluation rather than trying to quantitatively emulate a precise posture or motor action in simulation (this was stated in the methods, see line 740). This is now clarified in the Results as well (line 422).

5. *Some mention of what is already known about C. elegans salt learning should be made, so that the reader could place this work in the appropriate context.*

Reply: We have added some explanation on the behavioral switch studied (lines 20 and 21), but as the abstract may only contain 150 words there is no further space to mention more of what is already known.

Introduction

1. *The intro lacks focus and cohesiveness and fails to provide the reader with a fair idea of what are the premises of this work and consequently which is the niche the authors wish to fill. As a result, the reader is not well prepared of what there is to follow and is not provided with baseline information. Consider, for example, answering the question: what is the scope of this paper and why? Half of the intro is a summary of the findings, but it does not serve its role to introduce the reader to the work.*

Reply: We have rewritten the introduction to better introduce the reader to the subject of our study and to motivate the question and the approach we have taken.

2. *Adaptive decision making: the authors might wish to build a case for the term here, as it relates to the dynamic aspect of things, which is partly the focus of the mathematical model.*

Reply: We have added a paragraph expanding on adaptive decision making, including references.

3. Consider avoiding two antithetical sentences back to back in the first paragraph.

Reply: We have reworded these sentences.

4. “decisions are generally thought to be the result of integration of information from sensory neurons in downstream interneurons”: yes, and also given the animal’s prior experience.

Reply: We have added “prior experience” to this sentence.

5. “In the absence of a reward, the expected value of an action will wane”: the absence of a punishment has an analogous role in decision making.

Reply: We agree with the reviewer, but in the interest of brevity, we have removed this sentence.

6. “Like most animals, naïve *C. elegans* are attracted to NaCl”: you need to provide a citation for the “most animals” part, which I am not sure is the case, so generally speaking.

Reply: We agree that “Like most animals” is not very precise, and removed it.

7. It might be preferable to refer to [NaCl] or to NaCl concentrations instead of plain NaCl when authors write things like NaCl decrease or increase.

Reply: We have gone through the manuscript and corrected this point in all cases.

8. “ASEL is desensitized upon pre-exposure to NaCl; ASER is initially insensitive to NaCl changes and sensitizes upon pre-exposure; ASH nociceptive neurons become sensitized to considerably lower (non-toxic) levels of NaCl upon pre-exposure.”: I understand this summarizes many of the findings; however, it reads kind of confusing. Authors might want to avoid comparative terms (initially, lower: initially, meaning before what? Lower than what?) and restate the findings in a clearer and more enticing way.

Reply: We have reworded this paragraph.

9. The Weber law needs to be explained here. Its relevance to the findings or the hypothesis needs to be provided, too.

Reply: We added a sentence to explain the Weber-Fechner law.

10. “We show that a hierarchy of molecular, cellular and distributed circuit mechanisms are necessary and sufficient to fully capture our Ca²⁺ imaging results”: necessary and sufficient is a string statement. Authors need to reconnect with it and justify this claim later in the Discussion in a convincing way, if they wish to leave the claim here.

Reply: We removed the word “fully” and reworded the sentence. It now reads: “we identify a hierarchy of molecular, cellular and distributed circuit mechanisms that capture our Ca²⁺ imaging results”.

Results

1. I am not sure I follow the authors’ logic of which figures to present as main figures and which to include in the Suppl Information section. The opening part of the Results describes findings showcased in a Suppl Figure, which is weird, and it does not help the reader understand where is the focus of the paper, which information is of primary importance and which is accompanying information.

Reply: We had relegated “validations” (i.e. results already reported in similar form in the literature) and “negative results” (e.g. mutants with few effects) to supplementary results. We do agree with the reviewer that it would help the reader to start with a figure showing naïve responses in the ASE and ASH neurons. Therefore, we made the old Supplementary Figure 1, the new Fig 1.

2. “Naïve sensory responses”: it could also include that these are short term exposure responses, like the caption of Fig S1, see also headline of the next section.

Reply: We added “upon short-term exposure” to the caption.

3. *“but only rarely to lower concentrations”*: what does “rarely” mean?

Reply: We have reworded this sentence.

4. *“led us to conjecture that ASER responses”*: so is this a hypothesis? Do the authors hypothesize that? If so, then they can continue by saying “In order to test this hypothesis...” The manuscript could benefit significantly from stating the course of action in a hypothesis-driven way.

Reply: We added “To test this hypothesis” at the start of the next sentence.

5. *“We found that the propensity to respond, as well as the amplitude of the response increased with exposure time”*: the propensity to respond is hard to quantify, how do the authors achieve that? Please clarify.

Reply: With “propensity to respond” we meant the “fraction of animals that responded”. We reworded the sentence accordingly.

6. *Supplementary Table 1: Nice idea to provide all this information in one place. However, the table is not as helpful as it could be. Please provide the Fig number next to the reported response/neuron/experiment. Provide these data for all panels of all figures.*

Reply: We added the Fig. numbers to Supplementary Table 1.

I suppose there are “censored” animals, ie the ones that did not respond. Could this constitute some kind of “response index”? Could this be important? Does a “response index” or percentage regarding Ca transients correspond to the indices of behavioral assays?

Reply: We agree with the reviewer that depicting the fraction of animals responding as a response index is clearer than showing the number of animals responding. We have indicated the response index in addition to the total number of animals tested in Supplementary Table 1.

Importantly: Why is the rise time (mean and range) missing for some of the responses?

Reply: We measured the rise times for ASEL and ASH (naïve) and ASER (pre-exposed) animals as these struck us as being very different and we hypothesized that this difference could be important for the different roles that these neurons play in the behavior. We did not see clear differences between the rise times of naïve or pre-exposed ASEL or ASH neurons, therefore we did not include them. We have now included these data.

ASEL with pre-exposure: why are so few animals tested here, compared to the rest of the experiments, eg ASEL no pre-exposure?

Reply: We agree that relatively few animals were tested for the effect of the duration of the pre-exposure or the wash on the subsequent response in the ASEL neuron. However, we obtained reproducible results, showing that 5 minutes pre-exposure already results in significant desensitization of ASEL, which is even stronger after 10 minutes, and that only a 60 s wash is allowed between pre-exposure and the test exposure to maintain desensitization of ASEL.

In the caption, it reads “ASER stimulus steps are given in pre-exposure time”: this can be confusing. Please insert new columns/rows if needed, but please provide the info in a more straightforward way. This table is the identity of the experiments, so it needs to be well written.

Reply: We have included a column as suggested.

Regarding the ASH experiments (I did not try that for the rest): Exposure to 200mM [NaCl], no preexposure 7/18 animals respond 39%; preexposure 600sec 16/17 94% animals respond. Similarly, for 300mM: 68% vs 90% of animals; for 500mM the %s are similar. How do the authors comment on this? Is this important in their opinion?

Reply: The fact that more animals show ASH responses to 200 mM after pre-exposure is indeed part of the sensitization. We are confident about these numbers as they represent a very strong increase and correlate with the increased amplitudes of the responses. We don't think the smaller increase in the number of animals responding to 300 mM NaCl upon pre-exposure to 100 mM NaCl is significant (although these data do not allow a statistical analysis) as also the response amplitude did not change. We have included these results in the text.

7. Suppl Table 1, Fig 1, and in general: I think that a computation of the rising slope would help a lot with data analysis. Even more than rising time, mean and range, does. The slope can be very informative of the dynamic behavior of the system (neuron). Moreover, it might make a difference if used instead of the rise time to the max amplitude, like the authors do below (I have a comment on that later). The slope reflects how fast the Ca channels respond on the neuron's membrane.

Reply: We agree with the reviewer that computation of the rising slope would be useful. However, we are not convinced that our data and knowledge of the system is sufficient to allow such analysis. We think that more detailed calcium modeling (of buffering effects etc.) and perhaps better Ca²⁺ sensors (e.g. GCaMP6, integrated in the genome for stable and perhaps lower expression) would make such analyses more robust. Therefore, we did not compute the rising slopes of the responses of the neurons.

8. As mentioned before, the practice of discussing the results of >half the panels of Fig 1 after Fig 4 is discussed, is very confusing.

Reply: We agree. We made the old Fig 1a, b into Fig 2a,b and moved Fig 1c,d,e to a new Fig 5.

9. Fig 1: Is there a graph to show how wild type ASER responds to exposure following pre-exposure? Like Figs 1c, d, e?

Reply: We have included these data in the new Fig. 5.

10. What is the n for 1d and 1e?

Reply: We thank the reviewer for spotting this omission. We have now included these data in the legend of Fig. 5.

11. "To determine whether ASEL responses are also modulated by pre-exposure, we exposed animals to 100 mM NaCl for 60 seconds to 10 minutes and, after a 30 - 60 seconds wash, tested their response to 100 mM NaCl": I think 30 sec wash is not shown in Fig or in Table.

Reply: This is correct. We did not show the 30 sec wash, as they were not different from the 60 sec wash. We corrected the sentence.

12. "Hints of opposite forms of adaptation in ASER and ASEL were already reported by Oda et al., who found weak adaptation after 10 minutes of exposure to 20 mM NaCl19.": I think this might be better off in the Discussion.

Reply: We agree with the reviewer and moved this sentence to the Discussion.

13. Supplementary Fig. 3: How many animals tested (n)? What is the standard deviation or SEM? Is the chemotaxis index obtained in 50mM significantly different than the one for 10mM?

Reply: We have now indicated how often the assays were performed (n), what the error bars represent (SEM) and significance of differences. This figure is now Supplementary Figure 2.

14. Fig. 2: Have the authors conducted a time course study for ASEL, like they did for ASER? Why not 30 sec exposure for ASEL, to compare with ASER?

Reply: We conducted a time course study for ASEL, in that we varied pre-exposure time from 60-600 sec. and the length of the wash between pre-exposure and test-exposure. As ASEL responds to an increase in NaCl concentration, where Ca^{2+} concentrations start to drop after approximately 2 seconds, we did not vary the length of exposure, but only tested 3 sec exposure. Suzuki et al. (2008) exposed animals for longer time periods, and also report short lived Ca^{2+} fluxes in ASEL.

15. “We conclude that the ASEL response to NaCl desensitizes with pre-exposure and recovers in the absence of NaCl exposure, and conjecture that this sensory adaptation modulates the strength of attraction to NaCl in behavioral assays.”: First of all, I think that it is the neuron that sensitizes or desensitizes, not the response. Second, maybe this passage could fit better in the Discussion section. Instead of conjecture, would “suggest” be a good alternative here?

Reply: We agree that it is the neuron that desensitizes, although it is likely that the neuron is not completely desensitized and can still respond to other cues. We replaced “response to NaCl” with “neuron”. In addition, we replaced “conjecture” with “suggest”.

16. “We found response rates, amplitudes and profiles comparable to naïve responses to 500 mM NaCl”: Have the authors conducted some statistics to support that? How are these features, ie “profiles” comparable? “Thus, ASH neurons are sensitized by pre- exposure to NaCl.”: This too general a statement, maybe the authors would like make it more precise, so that it matches their findings.

Reply: With “profiles” we meant the overall appearance of the response. We agree this is not very clear and have therefore removed this phrase. Statistical analysis is displayed in Fig. 4c. We have specified the statement that the ASH neurons are sensitized.

17. “is prevalent in many nervous systems, including *C. elegans*21.”: *C. elegans* is not a nervous system, please be careful with wording.

Reply: We corrected this statement.

18. Authors should provide a justification for i) the neurons they are interested in, and ii) the mutants they have selected. Citations are good, but a proper justification needs to be provided some place, preferably the Results (alternatively the Discussion).

Reply: We elaborated on the choice of the mutants tested for their effect on desensitization of ASEL.

19. Why is Suppl Fig 4 a Supplementary figure? Why do authors provide only part of the Ca transient in Suppl Fig 4a (*odr-3*), c (*egl-3*)? What happens for the rest of the 600 sec?

Reply: We agree that it would be nice to also show this figure as a normal figure in the main text. We do, however, have to choose for only a subset of figures to present in the manuscript as there are limitations to the format of Communications Biology. Since none of the mutants tested showed a clear phenotype, we think this data can be presented as a supplementary figure (now Supplementary Figure 3). The responses of the animals were imaged for approximately 7-12 seconds, to prevent too much bleaching. In some cases, as for *odr-3* and *egl-3*, imaging lasted only for 7.5 seconds, explaining why these seem cut short. It would probably be interesting to follow these responses for a longer time, but because of bleaching that is not possible with our current set up.

20. “five out of nine animals tested showed a weak response to 100 mM NaCl after 10 minutes preexposure”: can the authors provide that data? It might be important for the rest of the argument.

Reply: See 21.

21. "Culturing *tph-1* animals for 72 hours on plates containing 2 mM serotonin resulted in complete desensitization of ASEL in 7 out of 9 *tph-1* animals tested": this is very interesting trend, but the data is few (compared to 5 out of 9 of the other case), the sample size is small; if 1 or 2 worms had behaved differently, the final conclusion would be different, but the sample size is small. I suggest the authors do not rely on these data this much, can the difference be evaluated by a statistical test? In relation to this, "hinting at an additional serotonin-dependent desensitization mechanism.": hinting is ok, but I think the relation to serotonin should not be based on the serotonin enriched plates data (see above).

Reply (20 and 21 together): As suggested by the reviewer, we re-analyzed the data of the ASEL responses of *tph-1* mutant animals focusing on statistics. This analysis showed that, although we did find some responses in *tph-1* animals after pre-exposure, the average maximum ratio change in pre-exposed *tph-1* animals was not statistically different from that of wt animals. We therefore rephrased our conclusions and removed the results of the experiments where we tried to rescue the *tph-1* defect by culturing animals on serotonin.

22. Moreover, the whole story of cell-autonomous response, I think the data really alludes to that, but it would take more mutants or even neuronal ablation of upstream neurons to make such a claim. Maybe the authors can re-state the claim? Besides, the serotonin data might be actually pointing the other way, if we had more data points we could tell with less uncertainty.

Reply: We have rephrased our conclusion to: "Taken together, our results suggest cell-autonomous desensitization of the ASEL neuron after prolonged exposure to NaCl.", and also changed the subtitle of this paragraph and wording elsewhere accordingly.

23. Related to that: are the authors certain that the strains they got have been outcrossed? And outcrossed sufficient times? Maybe the different responses among worms of the same strain are related to background mutations, very common if the strains are not backcrossed. For example, *odr-3(n1605)* is not outcrossed (see wormbase); *unc-13(e51)* I is not outcrossed, either. (Related to that, the authors should provide the strains names in the Methods section).

Reply: Almost all strains have been outcrossed, we have now indicated this, with the strain names, in the Methods section. Most imaging strains have been generated by crossing in the YC transgene. In some cases (*che-1* and *gpc-1*), the transgene was injected directly in the mutant background. In these particular cases, the involvement of *che-1* and *gpc-1* was previously confirmed by rescue experiments (Hukema et al. 2006). However, to prove that indeed a particular gene (e.g. *unc-13*) is required for sensitization of ASH, additional experiments are required, such as rescue or the analysis of a second allele. We added a remark to the conclusions of the section on sensitization of ASH.

24. Fig. 3, what is the n for all the panels?

Reply: Fig 3 has been split up in Figs 4 and 6. The n has been indicated in the legend.

25. "Transient pulse-like responses are well captured by two opposing and timescale separated components." Please provide citations of other papers that report similar findings.

Reply: We have included a reference.

26. "are gap-junctionally coupled": unusual selection of adjective instead of a noun here.

Reply: We have changed this remark to "are electrically coupled".

27. "Hyperosmotic responses were not considered": I guess for simplicity, but the authors need to justify that.

Reply: We reworded this to "As hyperosmotic responses are unaffected by gustatory plasticity, they were not considered in our model."

28. Fig 4: please consider changing the combination of black font on blue background wherever applicable.

Reply: We have changed the black font on dark backgrounds to white.

29. *“We parameterized the model using Ca²⁺ imaging data from fully sensitized ASEL and ASER neurons. Both the rise and decay times of the responses were consistently and significantly faster in ASEL than in ASER, consistent with previous work⁶”: this is interesting, because nowhere in the experimental results is the decay time quantified/calculated. I suggest the authors do that, otherwise the comparison is not supported.*

Reply: To support our comparison, we estimated decay time for ASEL and included this in the text (back to baseline in 3.9 ± 1.3 sec.). For ASER we cannot estimate the decay time to baseline, as the animals were not imaged long enough. We indicated that we observed a decrease of approximately 15% in 6 seconds.

“the ASER decay time is much slower than that of ASEL.”: again, I could find no data for this. Moreover, why is this repeated twice? Do the authors need to emphasize on something?

Reply: We combined the two sentences to remove the repetition.

Importantly: the amplitude is different in ASEL and ASER responses, indeed. Until ASEL max, though, the slope seems to be the same (they completely overlap in Suppl Fig 6). Could this mean that the dynamics/kinetics of the channels are the same, only the max amplitude differs? This might be important. Relevant to that: is the initial baseline similar/different?

Reply: We agree that the slopes of the initial responses in ASEL and ASER are similar. However, we feel we cannot compare amplitudes between different cells, because baseline Ca levels and YC expression levels are likely different.

30. Fig 4 is not a results figure; it is more like a methods/approach figure. Should another figure be used to report the modelling results? Maybe fig M3?

Reply: For modelers, using empirical results to construct a mathematical model is a result. This depicts a schematic of a mathematical model that we actually use to simulate the behavior. We have now merged this figure with the figure depicting the CI results of the model, resulting in a new Fig. 7.

31. *“consistent with the absence of a response in the naïve context”: where is this shown?*

Reply: This is shown in Fig. 1c. We now refer to this figure in the text.

32. Fig. 5 (now 7): what does the strikethrough indicate?

Reply: We have indicated in the legend of the figure what the strikethrough indicates.

“(a) Our model shows no difference in attraction in the quadrant assay between wild type and animals with always fully sensitized ASEL and ASER”: have the results of these animals been pooled? Why are there two bars in the graph?

Moreover, “wild type” is usually juxtaposed to mutants. There is confusion when wild type is juxtaposed to (de)sensitized, as I think is the case here, since these worms are “genetically” wild type. Maybe pick a different term?

Reply: We have clarified what we mean in the legend.

“(b) Ablating the synaptic connection from ASEL to the steering motor neuron or from ASER to the pirouette neuron significantly reduced the chemotaxis index.”: which model parameters correspond to that?

Reply: It's the synaptic weights between the respective sensory neuron and the motor circuits. We have rephrased this. We refer to the supplementary materials (and **Table M1**. Neuronal parameters and synaptic weights) for model details.

“(c) Double mutants with ablated ASEL and no steering, or ablated ASER and no pirouettes”: double mutants is a weird term for model animals, maybe the use of quotes would help here.

Reply: We have added quotes.

“(d) Experimental validation: animals with genetically ablated ASEL (OH8585) and genetically ablated ASER (OH8593) exhibit a strong reduction in the chemotaxis index in the quadrant assay”: the authors here name the strains, which they do not do (although they should) for the rest of the strains in Methods. Importantly: since the authors have these ablated strains, have they tracked them using a worm tracker, to see whether their locomotion (ie pirouetting, steering) differs from wild type (intact) animals? I think this would help support their claims and validate further their model.

Reply: We have added strain names to the Methods section.

We agree it would be interesting to test the animals in which one of the ASE neurons has been inactivated in our behavioral assay and track their behavior. However, the model that we present in this manuscript replicates the outcome of the assays that we do, the chemotaxis index, and not the physical behavior of an individual animal (though we have attempted to make our model animals behave as realistic as possible for a point worm). Therefore, a careful description of the behavior of individual animals does not fit in our manuscript. In addition, tracking animals (wild type and mutants) in our behavioral assay and detailed analysis of the behavior of individual animals posed technical challenges that we cannot solve with our current set up, especially in the current situation with limited time in the lab.

33. *“A sensitivity analysis on the key sensory parameters (ASEL/R rise and decay times) showed that, across the entire parameter space, introducing ASE adaptation led to significantly lower or at best equal chemotaxis (data not shown).”:* actually, a sensitivity analysis would be very interesting to see. Plus, what if you tweak the sensitization parameters or the [NaCl] value?

Reply: A sensitivity analysis over all relevant parameters would take up too much space in an already constrained manuscript. However, we agree with the reviewer that this point is crucial. In fact, by revisiting the question, we find that with the model in its final form, (de)sensitization does not affect model robustness at all! We have therefore changed this paragraph to state the new methods and results, added a section to the Supplementary Methods, and added a figure (Supp. Fig. 7a) to show our results. Specifically, we considered that the best way to probe the robustness of sensory adaptation, is to vary all relevant ASEL/R parameters at once, independently. We generated populations of model worms with different ASEL/R rise and decay times (sampled from a distribution with increasing “noise” amplitude around the default parameter values). We then compared the CI (on the quadrant assay) between populations of animals with and without (de)sensitization. We expected animals that lack sensory adaptation to perform better, but instead, we found that model animals with (de)sensitization are just as robust (achieving as high or marginally higher CI) as control animals subject to the same level of parameter variability. We further show that this result applies even for quite a large noise amplitude. For really high noise amplitudes, model animals are no longer able to perform the task as effectively, leading to a drop in the CI, but even in this regime, this degradation applies to both our test and control populations. We thank the reviewer for pushing us to revisit this point.

34. *In ASEL and ASER ablated animals, how do the Ca transients change, in ASH and in non-ablated ASE neurons?*

Reply: In animals in which the ASE neurons are absent (*che-1* mutant animals) naïve Ca transients in ASH are similar to wt, but we found no sensitization of the ASH neurons in *che-1* animals. These data have been presented in Figure 6c-e. We did not ablate ASEL or R specifically and test the Ca transients in ASH. We did test Ca transients in ASEL in animals that overexpress *Isy-6* in which the ASER neuron has been transformed to an ASEL neuron. These animals show wt ASEL responses (supplementary Fig. 3c; *otIs204*). ASH responses were not analyzed in these animals.

35. *“Without any further parameter tuning, simulations of naïve and pre-exposed animals closely reproduced these experimental data”: this is a strength and should be highlighted.*

Reply: We added “strikingly,”

36. *First paragraph of page 17: I think there is repetition here, and in addition maybe something is missing.*

Reply: Thank you for pointing this out. This is now fixed.

37. *I am not sure I understand the stochastic recruitment of ASH. Does this happen randomly? Can the authors elaborate on this, maybe in the discussion?*

Reply: We have rephrased the text in the results and discussion. Stochastic does not mean recruitment of ASH occurs randomly, but stochastic implies there exists a rate (or probability per unit time) that governs recruitment of ASH. This rate is experience-dependent.

38. *Fig. 7 (new Fig. 9): this is indeed a very interesting in silico experiment. I think it makes sense that they are ~1cm, ie ~1 worm body length, this is something I can see at the suppl video. In 7c, the change does not seem to be that much in all cases, similarly in 7d, although here a 30% change sounds more convincing. How big an area does this correspond to? How is this different than a no spot-containing area? Maybe this would help authors illustrate better their claim.*

Reply: The size of the arena is effectively infinite, and the statistics given after 120 minutes of simulation time. Each hop here corresponds to 3.3 cm (from one spot center to the next) and the worm size is 1 mm (though in our simulations they are reduced to point worms). We have indicated this in the legend. Indeed, the effect of adaptation is very significant (as seen in the movies). In Fig. 9c, we note the difference between ASH ‘ablated’ WT and no-adaptation is about 50% and in Fig. 9d, the difference between WT and no-adaptation is approximately 20%, but if ASH is ‘ablated’ the difference is over 6-fold. We have added this information in the caption.

The key in this set of simulation experiments is the difference between having adaptation and not having it. This will be lost in the case of no spots (because there is no salt, and hence no salt adaptation) so the simulation will only show random tumble-and-run trajectories with no bias. As there is no bias towards the spots the same statistics are not meaningful.

In addition, what is the max number of hops (to add to 7d)? this could also help highlight any differences. Or, how fast do they move from one hop to the next?

Reply: There is an endless number of statistics that could be generated from these movies. In Figure 7 (now 9) we included those that show the strongest effect of adaptation. By adding more statistics, the message will be diluted. For the reviewer's information, we include the simulation breakdown here.

# hops	WT	ASEL	ASER	adap	WT	ASEL	ASER	adap
1	8	15	11	11	13	17	0	86
2	73	76	57	79	52	76	40	176
3	152	154	125	141	156	185	120	147
4	128	155	144	154	135	132	172	67
5	92	77	105	72	102	73	104	17
6	44	18	38	32	35	15	55	7
7	2	4	18	9	6	2	7	0
8	1	1	2	2	1	0	2	0
9	0	0	0	0	0	0	0	0
mean+/ -sem	2.736+/ -0.056	2.556+/ -0.053	2.942+/ -0.060	2.678+/ -0.057	2.790+/ -0.056	2.442+/ -0.050	3.086+/ -0.053	1.548+/ -0.050

By watching the Suppl Videos 1-4, I was also wondering the following: How about the fraction of worms that leave the initial spot? And how about overall dispersal? In ASH- ASE (de)sens, are there secondary clusters formed or is it just mu idea? Lastly, in -ASH video, I think they move faster, can the speed be quantified? Also, what about the total area covered?

Reply: It is easy to "see" various patterns there, but we would like to emphasize that these patterns are all emergent predominantly from random reorientations whose rate is modulated by history of exposure. We would not read more into this. The locomotion speed of all worms across all movies is the same (0.1 mm/sec, see Fig M4), and the movies are all sped up by the same factor. We have double checked and find that the dispersal rate (i.e. growing radius of the outer 'ring' of the worms) is the same in videos 1 and 2.

39. How do the authors discuss their findings in the context of Klein et al, 2017 (doi: 10.7554/eLife.30503, Samuel and Karplus groups) and Calhoun et al, 2014 (doi.org/10.7554/eLife.04220.001, Chalasani and Sharpee groups)? No salt chemotaxis here, but exploration behavior.

Reply: The Klein et al. 2017 paper describes locomotion trajectories as a superposition of short-range deterministic motion and long-range diffusive motion. We are unable to compare these results empirically since we only zoomed in on a very small area, and otherwise, only counted worms in different quadrants. However, this description is consistent with our model assumptions. Because we did not quantify similar metrics, we feel it would be inappropriate to make any claims about this.

The paper by Calhoun et al. proposes an abstract drift-diffusion statistical model and suggests a hypothetical minimal 3-unit circuit (sensory, modulation, integration/decision making). In fact, depending on the exact formulation of such models, it is easy to see that the modulation can be implemented by distinct circuits or neurons, or else it can equivalently be incorporated into the existing units of the circuit as we do here. Here, we provide physiological and behavioral evidence for a particular form of modulation that is incorporated into the sensors and sensory circuits. We agree it would be interesting to discuss this more extensively, but we are very short on space.

Discussion

1. *Fig 8 and discussion in general: I am confused about the absence /presence of food. Do authors incorporate data from the literature in the schematic of Fig 8? Because as far as I can understand, the presented experiments do not explore both conditions.*

Reply: We did incorporate data from the literature in Figure 10 (old fig 8), which we discuss in the introduction. We have included a sentence to remind the reader of this fact.

2. *In relation to that: “our simulations demonstrate the feasibility of a parsimonious model in which recruitment of ASH by a food and an ASE (NaCl) derived signal underpin the switch between attraction and avoidance in gustatory plasticity”: where is the food signal in the model?*

Reply: It is unclear whether an absence of food signal recruits ASH, or whether presence of food inhibits recruitment of ASH. We have now indicated this in the legend of Figure 10 and added an arrow indicating that ASH is recruited in the absence of food.

3. *“C. elegans has seen a shift of computation from the inter- to sensory layers over its evolutionary history”: this is mentioned in lino, 2013, Salt Chemotaxis Learning in Caenorhabditis elegans, doi.org/10.1016/B978-0-12-415823-8.00013-7, Handbook of Neuroscience: “learning is in the sensory neurons”. How is this claim novel here?*

Reply: The review states that “The studies described in this chapter suggest that much of the learning and memory processes may be carried out in the sensory neurons.” In 2013 we knew the sensory neurons are important, but we did not know yet what happens in the sensory neurons. Here, we provide evidence of sensitization and desensitization of the ASE and ASH neurons and use modeling to show the importance of these processes in the behavioral switch. Our conclusions indeed fit with the suggestion of lino, but our claim is novel in the sense that we provide a firm basis for our conclusion.

4. *In general: I appreciate authors’ effort to keep discussion short, nevertheless I would like to see the results discussed more extensively in the context of similar findings and of C. elegans ecology, evolution and environmental needs. A short paragraph with basic conclusions/contributions to close the Discussion would also help.*

Reply: We have extended the discussion as suggested by the reviewer.

5. *How do the authors discuss their results in light of work done by Liu et al, 2018 (doi: 10.7554/eLife.36419, Leifer Lab) on context dependency of C. elegans response to mechanosensation?*

How do the authors discuss their results in light of the paper by Jang et al, 2019 (https://doi.org/10.1073/pnas.1821716116), lino Group, on starvation-dependent aversive navigation in Caenorhabditis elegans, which includes results on ASER?

Reply: The paper by Liu et al (2018) shows that animals respond differently to stimuli with different temporal properties, but focuses on behavioral output, rather than neural mechanisms. As we are very short on space, we have not added this reference. We added Jang et al 2019 to the discussion.

Methods

1. *Where animals synchronized? What age were the worms tested? “Adult” is a little vague.*

Reply: We added that animals were synchronized by bleaching and grown for 66-72 hours.

2. *What does “if possible” mean, regarding the use of glycerol to regulate osmolarity? In behavioral experiments: how many worms per plate? How many repetitions? Any censored animals?*

Reply: “if possible” means that in some cases the osmolarity of the stimulus buffer was too high. We changed “if possible” to “except when NaCl concentrations were too high”.

We tested approximately 100 animals per assay plate. This has been added.

Assays were performed in duplicate at least on two different days.

Animals that did not move away from the center or were located above the plastic edge were censored.

3. *“Assay duration ranged from 10 to 60 min but was 10 min unless specified otherwise”: can the authors explain the range in mins?*

Reply: Changed to: Assay duration was 10 minutes except in the experiments presented in Fig. 9, where animals were followed for 60 minutes.

4. *“Model parameters were set so the pattern generation was achieved endogenously, but the same model circuit would support alternative (proprioceptively driven) control mechanisms²¹”: why is this mentioned here? To show versatility of the model?*

Reply: We mentioned this here to show that the model is agnostic to the locomotion pattern generation mechanism. In other words, by including this statement we ensure that the readers would not assume that the specific mechanism used here is a model requirement or a model prediction.

5. *Have the authors considered trying a simulation with ASEL and ASH?*

Reply: All these combinations were tried. This one corresponds to ASER (virtual) ablation.

6. *Could the authors provide a figure with experimental trajectories (quadrant assay) and simulation trajectories? It is mentioned that adjustments were made so that simulation trajectories match experimental ones, so it would be useful to see a few.*

Reply: As discussed above the model that we present in this manuscript replicates the outcome of the assays that we do, the chemotaxis index, and not the physical behavior of an individual animal. Therefore, a careful description of the behavior of individual animals does not fit in our manuscript. In addition, tracking animals (wild type and mutants) in our behavioral assay and detailed analysis of the behavior of individual animals posed technical challenges that we cannot solve with our current set up, especially in the current situation with limited time in the lab. To answer the comment raised by the reviewer, we have quantified the number of pirouettes that animals make in the quadrant assay and included this number in the Methods.

Supplementary Methods

Computational Method

1. *I think a schematic of the model circuit would help.*

Reply: The full circuit is given in Fig M1. We are unsure what is missing in the eyes of the reviewer.

2. *Figure M1: is the DMN to DM connection really inhibitory?*

Reply: This is mathematically equivalent to changing the inhibition to excitatory and then taking the difference between DM and VM before theta output. We have clarified the figure accordingly.

3. *Can the authors elaborate (last paragraph of page 12) on how Ca transients data has shaped modeled sensory neurons?*

Reply: We have added a description on how this was done.

4. *Tables with parameters values: are all values determined in this work or are some based on values from the literature?*

Reply: Neuronal parameters are now explained on page 11 and 12. The speed of the worm and pirouette rate for the spot assay are set based on the literature (with citations given).

5. *Fig. M3b: is there an experimental current available from ASEL, to superimpose, as is the case for the other two neurons?*

Reply: Yes, we have added an experimental Ca flux to Fig. M3b.

6. *The Fechner's law appears to be quite central to this work, as it appears in various (at least 3) places in the manuscript. Would the authors consider devoting a paragraph somewhere to state the law and expand a little on its implications for organisms, providing a few examples? I think it would help with the overall narrative.*

Reply: We have slightly expanded the description of the Weber-Fechner law in the manuscript and refer the reader to the supplementary methods. Unfortunately, there is no space to expand on it in a separate paragraph.

7. *ASH recruitment: are there any examples in the literature of other neurons that behave similarly? Or other similar cases, more general?*

Reply: Yes, there are two examples that we refer to, Leinwand & Chalasani 2013 and Ghosh et al 2016. It is not always called recruitment, sometimes remodeling. See also Cohen and Sanders 2014 for a more in depth discussion.

8. *Parameter κ is the common rate parameter for recruitment kinetics; can the authors provide a citation?*

Reply: We are not sure what is meant by the reviewer. The equations are novel and the parameter value has been determined in this work based on the timescale of ASH sensitization and recovery.

9. *Suppl Video S5: is this peak concentration 100mM? Then why is the color bar set with max at 200 (blue) and the spots colored blue? Similarly, is Suppl Video S9 with peak concentration 200mM? Then why is the color bar the same with S1 (max 100mM)? S13 seems ok, on the other hand.*

Also, I think in S5 the dispersion distance between spots is not 5.0cm (compare to S13, for example). Is there any possibility that the videos have been mixed up? Please double check the sets of videos.

Reply: Thank you for spotting this mistake. We have double checked the movies and now provide the correct description. These movies are now Supplementary movies S15-S30.

10. *Is it possible to provide simulations videos of the simulated quadrant assay?*

Reply: We have generated the requested videos, corresponding to the data that we present in Figure 8. They have been added as Supplementary Movies S1-S14.

Reviewer #2

Summary:

*Dekkers and Salfeder et al., conduct a thorough analysis of the neuronal activity and adaptation of the main NaCl sensory neurons ASEL, ASER and the nociceptive ASH neurons. These analyses extend previous reports on de-sensitization of the ASEL and sensitization of the ASER neuron following exposure to low NaCl, and reveal that ASH also exhibit sensitization to lower concentrations. The authors conduct a genetic screen that suggests that (de)sensitization of ASE neurons is cell-autonomous, while the sensitization of ASH neurons requires the ASE neurons and various neurotransmitters. Lastly, the authors use the information regarding the NaCl sensory circuit to devise a simplified model of circuits and behavior to generate a comprehensive simulation. The model uses the NaCl sensory circuit response and adaptation to guide NaCl chemotaxis behavior. The simulation suggests that time-scale differences between ASER and ASEL guide pirouettes and steering behaviors, respectively, and avoidance after salt exposure mainly mediated by ASH. Efficient NaCl chemotaxis in the simulated quadrant assay did not require gustatory plasticity, at initial time-point but an experimentally observed temporal decline in chemotaxis index did. Simulating the behavior in a more complex "spot" assay suggests that plasticity in ASE and ASH neurons plays an important role in balancing aversion and attraction to achieve an optimal balance between exploration and chemotaxis behavior in more complex environments. I think these are interesting data and modelling results that provide a consistent model about gustatory plasticity in *C. elegans*. I see a conceptual advance in showing that gustatory plasticity at least in part could be explained by changes in a network of sensory neurons. I would be very enthusiastic about publication if the concerns below can be addressed.*

Major points:

1) Does the simulation of the ASE neurons recapitulate their activity dynamics?

See Supplementary Figure M3.

In this figure the simulated ASE and ASH neuron activity is plotted. However, the activity of the experimental values of ASEL and ASER neurons is plotted only for one, fully sensitized condition (for example for the ASEL the 30s pre-exposure to 100mM NaCl followed by a 30 seconds wash). While the model was parameterized according to the fully sensitized Ca²⁺ experimental data, it is important to show it can re-capitulate the (de) sensitized states of these neurons.

To my understanding the relevant experimental data for ASEL exposure to 60,120,300 and 600 seconds of 100mM NaCl is present in Figure 2a.

Same for the different wash times, the simulated data is presented in Figure M3b, while the experimental data in Figure 2e. They should be also shown side-by-side.

From my observation they do not seem to fully match the simulated responses, this should be addressed.

Reply: We have now added the in vivo Ca responses, from Figures 2 and 3, for comparison. Please note that the model responses have been scaled to the amplitude of the strongest response, making a direct comparison of the model panels to the respective in vivo traces difficult.

2) One key point observation is the different effects of ASER and ASEL on behavior relate to the different time constants of their Ca²⁺ response. For example It is claimed that "... a significant timescale separation between the Ca²⁺ responses of fully sensitized ASEL and ASER, both in their rise and decay times (Supplementary Fig. 6)." and also regarding the consequences on the model performance in simulated behavioral assays " Therefore, to effectively modulate this rate requires a memory of salt exposure over commensurate (or longer) timescales. The fast decay time of ASEL precludes this, while the slow decay time of ASER is ideally suited to modulate the pirouette rate effectively. " This is based on the result presented in supplementary figure 6. And a result from a previous study (H Suzuki and TR

Thiele et al., et al Neuron 2008, Figure 1).

While the slower decay response of the ASER neuron is convincing, I am not convinced that the rise of the ASER neuron is faster as claimed. As this plays an important role in the model and its interpretation, I think it would be beneficial to show how the results of the model change when the rise and/or decay rates of ASER and ASEL are equalized.

Reply: We agree with the reviewer (as also discussed in the response to Reviewer 1) that the difference in rise time between ASEL and ASER is less convincing than that of the decay time. Perhaps a better measure would be given by the rising slope. However, we are not convinced that our data and knowledge of the system is sufficient to allow such analysis, e.g. because of different Ca^{2+} levels in ASER and L and different expression levels of the Ca-imaging construct. We think that more detailed calcium modeling (of buffering effects etc.) and perhaps better Ca^{2+} sensors (e.g. GCaMP6, integrated in the genome for stable and perhaps lower expression) would make such analyses more robust. Using rise times that reflect the difference in rise time observed in our Ca^{2+} imaging experiments, we could reproduce the chemotaxis response of the animals (as measured in the chemotaxis index) of the quadrant assay in our model assay. In the absence of detailed modeling of the rise and decay of the underlying calcium signals, we were sure to add that if the rise time of ASER were shorter, the model predicts ASER would contribute to steering as well, and conversely, that if its decay time were shorter, a different (likely downstream) mechanism would be required to modulate the rate of pirouettes.

I am also curious to know how the authors view the different roles of ASER and ASEL in regulating runs and pirouettes with respect to the following previous publication (H Suzuki and TR Thiele et al., et al Neuron 2008) that "activation of ASEL elevated forward probability (Fig. 3c, ANOVA, $p < 0.01$) whereas activation of ASER had the opposite effect (Fig. 3d, ANOVA, $p < 0.05$). We conclude that ASEL activation causes runs whereas ASER activation causes turns; thus the functional asymmetry between ASE neurons extends to the level of behavioural output."

Reply: Our results nicely fit with the results reported by Suzuki et al. We now refer to this in the text.

Moreover, consistent with both models, the effect of genetic ablation of ASEL and ASER shows they are both required for NaCl chemotaxis. Analysis of pirouette and steering behaviors in these backgrounds during chemotaxis would allow the authors to validate their results and conclusions from the model, with respect to selective of ASEL and ASER in modulating these two behaviors.

Reply: We agree it would be interesting to test the animals in which one of the ASE neurons has been inactivated in our behavioral assay and track their behavior. However, tracking animals (wild type and mutants) in our behavioral assay and detailed analysis of the behavior of individual animals posed technical challenges that we cannot solve with our current set up. In addition, the current situation with limited time in the lab does not allow us to set up such experiments.

3) I think the following supporting data should be shown; it is important part of the logic of the paper. "A sensitivity analysis on the key sensory parameters (ASEL/R rise and decay times) showed that across the entire parameter space introducing ASE adaptation led to significantly lower or at best equal chemotaxis (data not shown). Therefore, a model that includes sensory adaptation is constrained to the small region of parameter space in which the detrimental effects of sensory adaptation are minimal if this model result reflects a true ... it begs the question what advantage is gained by sensory adaptation"

Reply: We agree that this is an important point. We now show that populations of animals with different values of rise and decay times achieve a higher chemotaxis index when ASE (de)sensitization is disabled, as compared to model animals that include (de)sensitization. We have weakened the phrasing of this sentence so it better matches this result. See also our reply to reviewer 1 on a similar point.

5) Regarding the statements on the role of serotonin in ASEL adaptation - "Only *tph-1* mutants showed slightly abnormal ASEL adaptation, suggesting some..." and later "while hinting at an additional serotonin-dependent desensitization mechanism" - The effect size shown for the *tph-1* mutants and serotonin Supplementary Fig 4c is very small and therefore I believe these statements should be moderated.

Reply: We re-analyzed the data of the ASEL responses of *tph-1* mutant animals. This showed that, although we did find some responses in *tph-1* animals after pre-exposure, the average maximum ratio change in pre-exposed *tph-1* animals was not statistically different from that of wt animals. We therefore rephrased our conclusions and removed the results of the experiments where we tried to rescue the *tph-1* defect by culturing animals on serotonin.

6) Regarding the finding in Figure 6b about the important role of ASH in plasticity induced avoidance, in comparison to the smaller role of ASE - validating these results with genetically or laser- ablated ASH animals, is needed to support their claims. Although, the authors refer to an earlier publication (Hukema 2006) on page 14, in Figure 3B of this publication gustatory plasticity defective in *odr-3* is not rescued by *odr-3* expression in ASH.

Reply: We have shown in that paper, Hukema et al. 2006, in Fig 2 that ASH is required for avoidance after pre-exposure: expression of *mec-2* and *mec-4d* in ASH strongly reduces avoidance. This confirms our claim and we now refer to this in the text.

The *odr-3* experiment indeed shows that *odr-3* functions in ADF in gustatory plasticity and probably not in ASH. The finding that expression of *odr-3* in ASH does not rescue gustatory plasticity does not show that ASH is not important, it suggests that *odr-3* does play a role in ASH in gustatory plasticity.

7) Page 17 Figure 7d and supplementary figure 9. "Enabling ASH recruitment further enhanced exploratory behavior (Figure 7b-d, supplementary fig 9)." Figure 7b-d does not show higher exploratory behavior (less or very similar # of hops and % of worms with three hops – comparing WT to ablated ASH WT). In contrast supplementary figure 9 shows higher exploratory for ablated ASH. The authors should explain these different results in the text, and in general clarify this part.

Reply: We thank the reviewer for spotting this. We have re-analyzed these data and found a mistake. Fig 9 (old Fig. 7) and supplementary Fig. 8 (old sup Fig 9) now show very similar effects. In addition, we have clarified the corresponding text.

8) The authors used a stochastic switch model for ASH recruitment. The rationale for this model design is unclear and not explained at all. Inspecting the numbers of non-responders vs responders in Suppl. table 1 does not justify stochastic recruitment of ASH versus ASE. Is this a plausible model? More explanations are needed here in results text and discussion.

Reply: We agree with the reviewer that ASH recruitment is unlikely to be as simple as an instantaneous and stochastic on-off switch. However, modeling the recruitment kinetics in greater detail would require sufficiently resolved kinetic data (on the fraction of responders as a function of time and concentration) which we lack. Our data showed a minority of responders to low concentration without pre-exposure, and a very high fraction of responders after 600 sec exposure. Without a clear indication of a change in the response amplitude, we opted for a parsimonious binary (on-off) model of ASH responsiveness (to low NaCl concentrations) and our model of the recruitment kinetics aimed only to capture the observed statistics with minimal assumptions. We agree that continuous models might have worked as well (for example mimicking the gain parameter of ASER), but did not feel the available data justified this. Modeling the recruitment of the ASH sensory neuron as a stochastic switch between on and off states sufficed for capturing the fractions of recruited ASH neurons on salt. Switching rates in this model vary with the animal's NaCl exposure history (a necessary assumption). In animals that have not been exposed to NaCl before, ASH is in its off or *unrecruited* state, in which it is implicitly assumed to respond only to

dangerous concentrations of NaCl (>300 mM). In animals that have been exposed to NaCl, ASH is in its 'on' or *recruited* state, and it responds to low concentrations of NaCl as well. We explain this model in detail in the Supplementary Methods. We believe that the reason this model suffices (for both the quadrant assay and spot assay) is because our simulations capture population behavior (chemotaxis index over time and spot hopping statistics, respectively), which relies on the fractions of animals that are recruited (as a function of time), and whether such recruitment happens on or off salt. More detailed models of ASH recruitment would require additional data and are therefore beyond the scope of this study.

9) *Throughout the paper, statistics are performed only for selective sets of experiments. For example, data shown in Figure 1c-e lack any statistics. This needs to be fixed throughout all experimental results.*

Reply: In the manuscript, we only showed the statistically significant differences. We have now included statistical analysis in the legends of the figures, to indicate non-significant differences.

10) *Comparing the n numbers in Suppl Fig 1a-b, it seems only responders were used here for quantification and plotting. What is the justification for this?*

Reply: In Sup Fig. 1, now Fig. 1, we show that ASEL responds to an increase in NaCl concentration, over a broad range of NaCl concentrations. Not all animals respond, as can be seen in Sup Table 1. We think that the fraction of animals that responds is an indication of the robustness of the response, how well this concentration is sensed by the animal. However, we cannot exclude that the animals do not respond due to technical reasons. Our quantifications could be strongly affected if we would include all animals, including those that did not respond. Therefore, we present these data separately: Fig 1 (old Sup Fig. 1) to show the average response of animals that do respond and sup Table 1 to show the fraction of animals that responded.

Minor points –

-(Optional) in discussion “To our knowledge, this would be the first example of sensory neurons encoding and controlling distinct motor actions” I am not sure that statement is correct.

Reply: The reviewer is absolutely correct and we should have credited Suzuki et al. for this important contribution. The key word in the above sentence (which was not sufficiently clear) is “encode”. Suzuki et al. beautifully showed that the activation of sensory neurons has a statistical behavioral effect (the probability/length of a run, versus the probability of a pirouette). But this is difficult to achieve in practice if the two neurons are only encoding the change in the NaCl concentration (with the ON neuron encoding positive changes and the OFF, negative changes). We propose an *encoding* of these behaviors through the kinetics of activation, and specifically through the time scales of the response. A fast rise time in ASEL allows the neuronal activation to control steering. A slow decay time in ASER allows the neuronal activation to control the rate of pirouettes. In such a picture, even if the downstream circuit were identical (as it is in our model), the behavioral consequences would be different. We have now clarified this in the text.

- Controls should be shown in the relevant figures, not in a separate figure to allow comparison.

This is relevant to the following figures:

-Figure 3 - a control of ASH response to 200mM NaCl in naive worms not pre-exposed to NaCl is missing to allow comparison (like the one shown in supplementary figure 1d).

-Supplementary figure 4 - the ASEL desensitization response in wild type (like the one in Figure 2c) should be shown again in this figure to allow a comparison.

Reply: We agree with the reviewer that including the controls would help comparison. We added the naive ASH response from Fig. 1 to Fig. 4 (previously Fig 3), for comparison. We also added the wt ASER responses from Fig. 2 to Fig. 5 and Sup. Fig. 4, the wt ASH

responses from Fig. 4 to Fig. 6, wt ASER responses from Figs. 1 and 3 to Sup. Fig. 3.

*-Page 9 under "Sensitization of ASER is also predominantly cell-autonomous"
"we found a strong response of the ASER neuron of unc-13(e51) animals.. (Fig 1c.)"
Is it ok to refer to figure 1c only so late at the paper after figure 3 has already been
resented?*

Reply: We agree with the reviewer that the way we arranged the figures was not ideal. In the original submission we chose to present the data per cell and show the naïve responses in the supplementary data, to reduce the number of figures. We have now adapted the figures as suggested by the reviewer.

- The phrasing of the legends of the figures is a bit messy, and the sentences can clarified better. For example: legend of Figure 1 (Page 5) "did not result in response, 10 minute of exposure did albeit 2 seconds later than in wild type" - this is a very lengthy sentence that is not very clear.

Reply: We have tried to clarify the legends.

*- The reference to the previous publication of Oda et al (J Neurophysiol 2011) "Hints of opposite forms of adaptation in ASER and ASEL were already reported by Oda et al., who found weak adaptation after 10 minutes of exposure to 20mM NaCl. "
The adaptation in Oda et al paper (Figure 1) is not weak
Other authors recognized that the effect is there citing Oda et al
For example Katz et al., Cell reports 2018: "Imaging of ASER reveals that salt cues usually induce calcium signals that drive synaptic vesicle release. Following starvation, however, neuronal outputs are inhibited even though calcium signals are enhanced (Oda et al, 2011)."
Beets et al., Worm, 2013: ". This is supported by the observed change in neuronal activity of ASE neurons following the short pre-exposure of worms to salt in the absence of food 55 . "*

Reply: Indeed, the adaptation reported by Oda et al are not weak. We have removed the word "weak".

Page 7 under "Prolonged exposure to NaCl sensitizes ASH" - a not clear phrasing of sentence "We found response rates, amplitudes and profiles comparable to naive responses to 500mM". This phrasing is a bit confusing. The important comparison here is not pre-exposed ASH versus pre-exposed ASH, when exposed to the same NaCl concentration of 200mM. The ASH neuron in naive worms is not responding while ASH in pre-exposed worms is responding. Therefore, I think it is better to compare the results of the sensitized ASH to the non-sensitized ASH and only then note that these responses were like the responses to high concentrations.

Reply: We agree with the reviewer and changed the phrasing accordingly.

-Figure 7b-d – since average of 1000 simulated worms is presented, it would be more informative to show violin plot or another representation that shows how the distribution of the values (perhaps there's a bi-modal distribution?)

Reply: We have now included the error bars in Fig. 9 (old Fig. 7). We present them as bars for consistency with the rest of the manuscript. As can also be seen from the size of the error bars, there is no bimodal distribution.

-Figure 6b. why ASH 100mM and 0mM ASH-on are not presented?

Reply: In Fig. 8b (previously 6b) we test the contribution of ASH to the response of animals pre-exposed to 0 or 100 mM NaCl. In animals pre-exposed to 0 mM NaCl, ASH has not been recruited, but becomes recruited in time on the quadrant plate. Therefore, we test what the response of the animals would be if ASH could not be recruited. Testing what happens if ASH would be recruited from the start, would not address what we aim to test and be distracting. Similarly, in animals pre-exposed to 100 mM NaCl ASH is recruited but desensitizes in time. Therefore, we tested what would happen if ASH remains recruited.

REVIEWERS' COMMENTS:

Reviewer #1 (Remarks to the Author):

The authors have successfully addressed most of my questions, both the ones asking for more clarification as well as the ones in which I was questioning some of their choices. In addition, the authors have added new data and analyses, and they provide more information about their experimental process and evaluation of results. The revised manuscript is in a much better shape than the original one, the flow is smoother, results are presented in a more intuitive and concise way, the findings are better contextualized, introduction and discussion sections are leaner and more to the point. In any case, even if some things could have been handled differently, the decision on how to present data, what story to tell, and which aspects of the findings to highlight lies with the authors. I believe that this work will be of significant value to the community, the findings of both the experiments and the mathematical effort are important, and I trust that many researchers will appreciate and benefit from this contribution.

I only have a handful of remaining questions/concerns, and I welcome publication of the revised manuscript after these issues are addressed:

1. Line 44-45 "Often a goal is implicit as the driver for adaptive behavior": I think I understand the meaning of the sentence, but could something be missing here? (This is what I understand that the authors want to say: "Often, the actual driver of an adaptive behavior might act in a less apparent way", or something along these lines. Maybe the authors can rephrase this sentence so that its meaning is more straightforward?)

2. Lines 324-330: The authors state that "Interestingly, all mutants tested, *unc-13(e51)*, *eat-4(ad819)*, *unc-31(e928)*, *egl-3(ok979)*, *cat-2(tm2261)* and *tph-1(mg280)*, showed reduced sensitization of ASH, but responded as wild type to 500 mM NaCl", referring to Fig 6f-k. In the caption of Fig 6, it is stated that "(g) Two out of nine *eat-4(ad819)* animals responded. (h) Three out of seven *unc-31(e928)* animals responded. (i) Three out of nine *egl-3(ok979)* animals responded. (j) Three out of nine *cat-2(tm2261)* animals responded. (k) One out of eight *tph-1(mg280)* animals showed sensitization of ASH." Therefore, in my understanding, when the authors say in the text that "all mutants tested [...] responded as wild type to 500 mM NaCl", they rely on the responses of 2/9 animals (6g), 3/7 animals (6h), 3/9 animals (6i), 3/9 animals (6j) and 1/8 animals (6k). As for the wild type, as mentioned in Fig 4, for 500 mM (preexposure), it is n=24, out of which 19 animals have responded (Suppl Table 1). So, although the magnitude of the mutants' response might be similar to the one of the wild types, the fact that so few mutant animals responded, in contrast to the wild type animals, does not allow much room for the claim that the mutants in this case respond "as wild type". If the authors can make clear that their statement refers to the magnitude (or some other trait) of the response and not its robustness, then it would be more accurate.

3. My previous comment #21: The authors say that they "removed the results of the experiments where we tried to rescue the *tph-1* defect by culturing animals on serotonin", based on a new analysis they run as part of the revision process. However, lines 331-338, pg. 15 of the revised manuscript are identical to the respective paragraph in pg. 10 of the original manuscript, except from the cited figure number. It is unclear to me what the authors intend to do with this set of results.

4. Line 718, and my previous comment #1 referring to Methods: The hrs is missing from line 718 (66-72 hrs).

5. My previous comment #10 on the Suppl Information section: the authors state that "We have generated the requested videos, corresponding to the data that we present in Figure 8. They have been added as Supplementary Movies S1-S14." I can see the description added in the Suppl Info pdf, but for some reason I cannot see the videos themselves. Have they been submitted? In fact, I cannot find any videos as part of the material submitted for revision.

Reviewer #2 (Remarks to the Author):

The authors have largely addressed my comments and the revised manuscript represents major improvements, also upon addressing the comments of the other reviewer. I think this is an important piece of work showing that plasticity in a network of sensory neurons likely serves an important ethological function. It paves way for interesting future work that can confirm and further optimize their model parameters and to also perform behavioral experiments in similar complex environments. I think at the current state this project has sufficient impact for publication.

Thank you for accepting our paper for publication in *Communications Biology*. We have addressed the remaining concerns of the reviewers and edited our manuscript according to your format requirements.

Below, please find our response to the comments of reviewer 1:

1. Line 44-45 "Often a goal is implicit as the driver for adaptive behavior": I think I understand the meaning of the sentence, but could something be missing here? (This is what I understand that the authors want to say: "Often, the actual driver of an adaptive behavior might act in a less apparent way", or something along these lines. Maybe the authors can rephrase this sentence so that its meaning is more straightforward?)

Reply: We have rephrased this sentence to "Often, the goal of a behavior is implicit (e.g. tracking an animal and recording its neural activity may not disclose what form of reward the animal seeks as it acts in its environment). Furthermore, the advantage gained by adaptive behavior may be similarly elusive."

2. Lines 324-330: The authors state that "Interestingly, all mutants tested, unc-13(e51), eat-4(ad819), unc-31(e928), egl-3(ok979), cat-2(tm2261) and tph-1(mg280), showed reduced sensitization of ASH, but responded as wild type to 500 mM NaCl", referring to Fig 6f-k. In the caption of Fig 6, it is stated that "(g) Two out of nine eat-4(ad819) animals responded. (h) Three out of seven unc-31(e928) animals responded. (i) Three out of nine egl-3(ok979) animals responded. (j) Three out of nine cat-2(tm2261) animals responded. (k) One out of eight tph-1(mg280) animals showed sensitization of ASH." Therefore, in my understanding, when the authors say in the text that "all mutants tested [...] responded as wild type to 500 mM NaCl", they rely on the responses of 2/9 animals (6g), 3/7 animals (6h), 3/9 animals (6i), 3/9 animals (6j) and 1/8 animals (6k). As for the wild type, as mentioned in Fig 4, for 500 mM (preexposure), it is n=24, out of which 19 animals have responded (Suppl Table 1). So, although the magnitude of the mutants' response might be similar to the one of the wild types, the fact that so few mutant animals responded, in contrast to the wild type animals, does not allow much room for the claim that the mutants in this case respond "as wild type". If the authors can make clear that their statement refers to the magnitude (or some other trait) of the response and not its robustness, then it would be more accurate.

Reply: This is indeed confusing. In all cases the responses to 500 mM NaCl were as in wild type. We mean with the 2/9 that responded that 2 animals responded to 200 mM NaCl of 9 animals that all responded to 500 mM NaCl. For wild type these numbers are 16 animals that responded to 200 mM NaCl of the 17 animals (see Supplementary Table 1). We have rephrased the legend to Fig. 6f-k to make this clearer.

3. My previous comment #21: The authors say that they "removed the results of the experiments where we tried to rescue the tph-1 defect by culturing animals on serotonin", based on a new analysis they run as part of the revision process. However, lines 331-338, pg. 15 of the revised manuscript are identical to the respective paragraph in pg. 10 of the original manuscript, except from the cited figure number. It is unclear to me what the authors intend to do with this set of results.

Reply: The previous comment #21 dealt with experiments where we tried to restore responses in ASE in *tph-1* mutant animals by culturing the animals on serotonin. These results were inconclusive and therefore removed from the revised manuscript. Here we tried to rescue responses in ASH, in *tph-1* animals by culturing the animals on serotonin. The rescue experiment in *tph-1* animals is also

inconclusive, but could be interesting because it shows that culturing wild type animals on serotonin and thus supplying an excess of serotonin also affects sensitization of ASH. However, as the meaning of these results are not so clear and confusing, we removed these from the manuscript.

4. Line 718, and my previous comment #1 referring to Methods: The hrs is missing from line 718 (66-72 hrs).

Reply: We have corrected this error and included “hours”.

5. My previous comment #10 on the Suppl Information section: the authors state that “We have generated the requested videos, corresponding to the data that we present in Figure 8. They have been added as Supplementary Movies S1-S14.” I can see the description added in the Suppl Info pdf, but for some reason I cannot see the videos themselves. Have they been submitted? In fact, I cannot find any videos as part of the material submitted for revision.

Reply: We are sorry for this omission. We have now included all movies with our submission.